# A natural bacterial pathogen of *C. elegans* uses a small RNA to induce transgenerational inheritance of learned avoidance

Titas Sengupta[1,2], Jonathan St. Ange[1,2], Rachel Kaletsky[1,2], Rebecca S. Moore[1,2], Renee J. Seto[1,2], Jacob Marogi[2], Cameron Myhrvold[2], Zemer Gitai[2], Coleen T. Murphy[1,2]*

**1** Lewis Sigler Institute for Integrative Genomics, Princeton University, Princeton, New Jersey, United States of America, **2** Department of Molecular Biology, Princeton University, Princeton, New Jersey, United States of America

* ctmurphy@princeton.edu

**Data Availability Statement:** Bacterial small RNA sequencing data are available at NCBI Bioproject PRJNA1062118. The numerical data for all the

## Abstract

*C. elegans* can learn to avoid pathogenic bacteria through several mechanisms, including bacterial small RNA-induced learned avoidance behavior, which can be inherited transgenerationally. Previously, we discovered that a small RNA from a clinical isolate of *Pseudomonas aeruginosa*, PA14, induces learned avoidance and transgenerational inheritance of that avoidance in *C. elegans*. *Pseudomonas aeruginosa* is an important human pathogen, and there are other *Pseudomonads* in *C. elegans'* natural habitat, but it is unclear whether *C. elegans* ever encounters PA14-like bacteria in the wild. Thus, it is not known if small RNAs from bacteria found in *C. elegans'* natural habitat can also regulate host behavior and produce heritable behavioral effects. Here we screened a set of wild habitat bacteria, and found that a pathogenic *Pseudomonas vranovensis* strain isolated from the *C. elegans* microbiota, GRb0427, regulates worm behavior: worms learn to avoid this pathogenic bacterium following exposure, and this learned avoidance is inherited for four generations. The learned response is entirely mediated by bacterially-produced small RNAs, which induce avoidance and transgenerational inheritance, providing further support that such mechanisms of learning and inheritance exist in the wild. We identified Pv1, a small RNA expressed in *P. vranovensis*, that has a 16-nucleotide match to an exon of the *C. elegans* gene *maco-1*. Pv1 is both necessary and sufficient to induce learned avoidance of Grb0427. However, Pv1 also results in avoidance of a beneficial microbiome strain, *P. mendocina*. Our findings suggest that bacterial small RNA-mediated regulation of host behavior and its transgenerational inheritance may be functional in *C. elegans'* natural environment, and that this potentially maladaptive response may favor reversal of the transgenerational memory after a few generations. Our data also suggest that different bacterial small RNA-mediated regulation systems evolved independently, but define shared molecular features of bacterial small RNAs that produce transgenerationally-inherited effects.

plots in all the figures have been included in S1 Data.

**Funding:** This work was supported by a Pioneer Award to CTM (NIGMS DP1GM119167), a Transformative R01 Award (1R01AT011963-01) to ZG & CTM., CDCP 75D30122C15113 to CM, T32GM007388 (NIGMS) support of RSM., a Damon Runyon Fellowship (DRG-2481-22) to TS, a Ford Foundation Predoctoral Fellowship to RS (https://www.nationalacademies.org/our-work/ford-foundation-fellowships), and an NSF GRFP (DGE-2039656) predoctoral award to JS. The funders had no role in study design, data collection and analysis, decision to publish, or preparation of the manuscript.

**Competing interests:** I have read the journal's policy and the authors of this manuscript have the following competing interests: ZG is the founder of ArrePath.

## Author summary

*C. elegans* can learn to avoid a pathogenic clinical isolate of *Pseudomonas aeruginosa*, PA14, for four generations after training, through ingestion and RNA interference processing of a bacterial small RNA, P11, that targets a *C. elegans* neuronal gene, *maco-1*, through a 17-nucleotide perfect match. We screened bacteria associated with *C. elegans* in the wild, and found that lab *C. elegans* as well as wild *C. elegans* strains can also learn (and remember) to avoid *P. vranovensis*, a wild *Pseudomonas* pathogen. *P. vranovensis* uses a different small RNA that we identified and named Pv1, which targets a different exon of *maco-1* (through a different 16-nucleotide match) and downregulates *maco-1* expression transgenerationally, resulting in transgenerational inheritance of learned *P. vranovensis* avoidance. These data suggest that this mechanism of learning and remembering pathogen avoidance likely happens in the wild. Furthermore, the similarity in the "smell" of pathogenic and nutritious *Pseudomonas* (*P. mendocina*) may exert evolutionary pressure to forget the learned avoidance by the fifth generation, to prevent the worms from missing out on good food sources while avoiding pathogens.

## Introduction

Plants and animals have evolved diverse mechanisms to adapt to constantly changing environmental stimuli. Some of these stimuli are encoded as molecular changes that do not involve changes in DNA sequence, but are instead epigenetic, that is, mediated through changes in non-coding RNAs, DNA modifications, histone modifications, and nucleosome positioning [1–6]. These changes can occasionally cross the germline and confer adaptive benefits to the first generation of progeny (intergenerational) [7–20] or more (transgenerational) [21–33]. Multigenerationally-inherited effects can provide adaptive advantages in changing environments, particularly in organisms with short generation times [34–36].

Over the past decade, instances of multigenerational inheritance have been reported in various organisms [37–39]. We previously characterized an example of epigenetic inheritance in response to a physiological stimulus, highlighting its adaptive benefits in *C. elegans*: upon exposure to the pathogenic *Pseudomonas aeruginosa* strain PA14, worms learn to subsequently avoid the bacteria, then pass on this learned avoidance to four generations of progeny [27]. A single small RNA from PA14, P11, mediates this avoidance and its transgenerational inheritance through downregulation of the worm neuronal gene *maco-1*, which results in a switch from attraction to avoidance behavior [23,26]. These studies provided the first example of bacterial small RNA-mediated regulation of a learned behavior and its transgenerational inheritance [23,26,27]. However, PA14 is a human clinical *Pseudomonas* isolate; whether bacteria in *C. elegans'* natural environment elicit learned responses and multi-generational inheritance of learned responses through small RNAs is not known.

Diverse bacterial species influence *C. elegans* physiology and life history traits [40–42]. Bacterial species from *C. elegans'* natural environment have been systematically characterized [43–49]. These studies revealed multiple features of the microbiota in *C. elegans* natural habitat and their relationship to host physiology [50,51]. Studying bacterial species that are naturally associated with *C. elegans* might reveal processes that occur in the wild, and not in the laboratory, and vice versa; for example, bacteria from *C. elegans'* natural environment suppress mortal germline phenotypes that wild worms exhibit on laboratory strains of *E. coli* [52]. Therefore, it is important to test the physiological relevance of laboratory experimental results under more natural conditions.

Bacterial species in the worm microbiome that induce stress and immune response reporters are categorized as pathogenic [47], while species that promote increased worm growth rates are categorized as beneficial [47,49,53], and some of these species confer protection against pathogenic species [44,54]. Other species are beneficial in some contexts and pathogenic in others [55]. Therefore, it may be evolutionarily favorable for worms to have plastic responses to different bacterial classes that they naturally encounter. Worms are naively attracted to specific beneficial and neutral bacterial species (e.g., *Pseudomonas mendocina* and *Proteus mirabilis*, respectively) when given a choice between these bacteria and their laboratory diet *E. coli* HB101 [56]. Similarly, worms grown on the beneficial bacterial strain *Providentia alcalifaciens* prefer this bacterial species over their laboratory diet *E. coli* OP50 in a behavioral choice assay [57]. These beneficial bacterial species modulate *C. elegans'* attraction towards several chemicals. Naïve or learned attraction in response to beneficial bacteria that worms encounter in their natural environment may have evolved as an evolutionarily favorable strategy. However, it is not known if worms can learn to avoid the various pathogenic bacterial species in their environment, or if they can inherit this learned avoidance. Additionally, whether bacteria in *C. elegans'* natural environment can modulate the host nervous system through small RNAs and whether they can induce transgenerationally inherited effects are not known.

*Pseudomonas* is one of the largest among the bacterial genera that constitute *C. elegans'* natural microbiome [47]. In this study, we examined *C. elegans'* behavioral responses to a Pseudomonad species present in its natural microbiome. We found that an isolate of *Pseudomonas vranovensis* can elicit learned avoidance and its transgenerational inheritance through a single small RNA that is both necessary and sufficient. However, this learned response to *P. vranovensis* also leads to avoidance of a beneficial bacteria also found in *C. elegans'* environment, *P. mendocina*. Our work reveals a transgenerational effect in response to bacteria in *C. elegans'* natural microbiome, underscoring the physiological relevance of transgenerational inheritance and its significance in the wild. We also identified a new small RNA that can induce a learned behavior in *C. elegans*, therefore expanding the repertoire of bacterial small RNA-mediated regulation of the host nervous system and helping to identify characteristics of small RNAs necessary for trans-kingdom signaling. Finally, the induced avoidance of a beneficial bacteria after pathogen training suggests that "forgetting" learned pathogen avoidance after a few generations might benefit *C. elegans*, limiting maladaptive behaviors.

## Results

### Wild microbiome bacteria induce learned avoidance

To examine if exposure to bacteria isolated from *C. elegans'* natural environment can produce stereotypic behavioral responses and further, whether these could potentially be small RNA-mediated, we tested *C. elegans'* response to strains that are present in its natural microbiome [47]. We chose nine different bacterial species, mostly from the CeMBio collection [50] to test. These include non-pathogenic bacteria that are beneficial, as they enhance worm growth rates or provide protection against pathogen infection, or have positive or neutral effects depending on the physiological context (*Pseudomonas mendocina* (MSPm1), *Raoultella sp.* (Jub38), *Leliottia sp.* (Jub66), and *Acinetobacter guillouiae* (Myb10), *Ochrobactrum vermis* (Myb71), *Enterobacter hormaechi* (CEN2ent1), as well as three bacteria that are pathogenic or impair worm growth and development (*Stenotrophomonas maltophilia* (Jub19), *Sphingobacterium multivorum* (Bigb0170), and *Pseudomonas vranovensis* (GRb0427) [46,47,49,50]. Starting as late L4 animals, we exposed *C. elegans* for 24hrs either to OP50 *E. coli* (the standard lab cultivation strain) or to the test bacteria, and then assayed their preference to OP50 vs. the test strain (**Fig 1A**). In general, *C. elegans* prefer the wild strains—both beneficial and pathogenic—

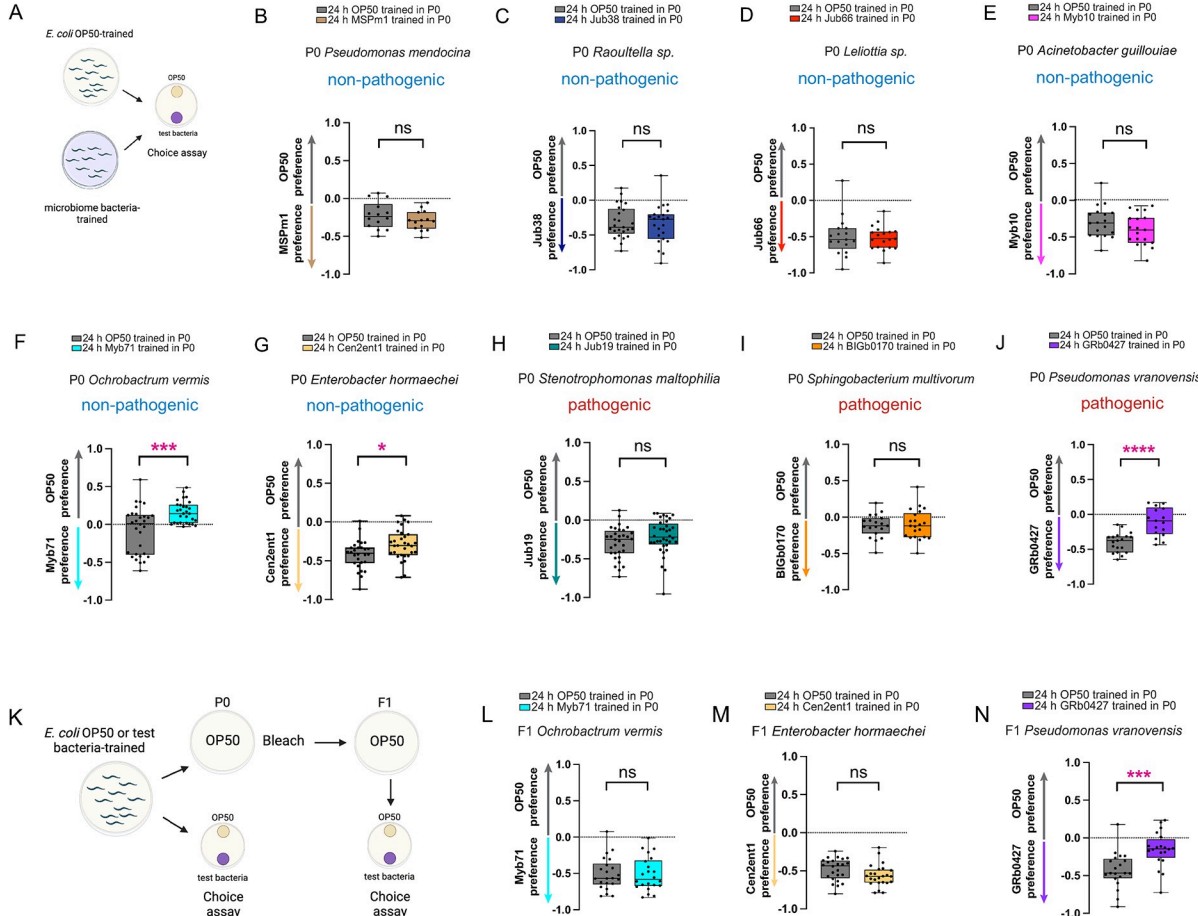

**Fig 1. Wild microbiome bacteria induce learned avoidance. (A)** Worms trained for 24 hours on *E. coli* OP50 or a test wild microbiome bacterial strain are tested in a choice assay between OP50 and the test bacterial strain. **(B-J)** Choice assays before and after training on *Pseudomonas mendocina* MSPm1 (B), *Raoultella sp.* Jub38 (C), *Leliottia sp.* Jub66 (D), *Acinetobacter guillouiae* Myb10 (E), *Ochrobactrum vermis* Myb71 (F), *Enterobacter hormaechei* CEN2ent1 (G), *Stenotrophomonas maltophilia* Jub19 (H), *Sphingobacterium multivorum* Bigb0170 (I), *Pseudomonas vranovensis* GRb0427 (J). **(K)** Worms trained for 24 hours on *E. coli* OP50 or a microbiome bacterial strain are bleached to obtain eggs, which are allowed to grow to Day 1 adults on OP50 plates. These adult F1 progeny are tested in a choice assay between OP50 and the respective bacterial strain. **(L-N)** Choice assays with F1 progeny of OP50 and Myb71-trained (L), OP50 and CEN2ent1-trained (M), and OP50 and GRb0427-trained (N) animals. Each dot represents an individual choice assay plate. Boxplots: center line, median; box range, 25th–75th percentiles; whiskers denote minimum-maximum values. Unpaired, two-tailed Student's t test, ****p < 0.0001, ***p < 0.001, and *p<0.05, ns, not significant. Schematic representation in (A) and (K) were created using Biorender.

relative to the laboratory food *E. coli* OP50 (OP50-trained (gray bars), **Fig 1A–1J**). After 24hr exposure to wild bacteria (training), six of the bacteria showed no significant change in preference, despite the fact that two of those strains, Jub19 and Bigb0170, have detrimental effects on worms [47,50]; however, 24hr of exposure to three strains (*Ochrobactrum vermis* (Myb71), *Enterobacter hormaechi* (CEN2ent1)), and *Pseudomonas vranovensis* (GRb0427)) induced significant avoidance in the trained mothers (P0) (**Fig 1F, 1G and 1J**). That is, upon cultivation on a bacterial lawn for 24 hours, worms learn to robustly avoid the bacteria, as shown in a choice assay between OP50 and the test strain. (This P0 learned avoidance has been previously reported for *Ochrobactrum vermis* (Myb71) [58]). Longer exposure (36 hr) to the detrimental bacteria *Sphingobacterium multivorum* (Bigb0170) does not induce avoidance (S1 Fig). Thus, it seems that most wild strains are inherently attractive to *C. elegans*, whether they are

beneficial, neutral or pathogenic, only a subset of strains induce avoidance in the P0, and this first-generation avoidance does not correlate with pathogenicity of the strain.

To determine whether this P0 learned avoidance is inherited by the next generation, trained mothers (P0) were bleached and their progeny (F1) were raised on OP50 *E. coli* until adulthood, then were tested for their choice (with no F1 training) (**Fig 1K**). We observed that although neither *Ochrobactrum vermis* (Myb71) nor *Enterobacter hormaechi* (CEN2ent1)) progeny inherited the learned avoidance from their mothers (**Fig 1L and 1M**), progeny of *Pseudomonas vranovensis* (GRb0427)-trained mothers also avoided *Pseudomonas vranovensis* (GRb0427) (**Fig 1N**). Of the various bacteria we tested, only the pathogenic *Pseudomonas vranovensis* (GRb0427) induces learned avoidance and inheritance of avoidance.

## *C. elegans* learn to avoid the natural bacterial pathogen, *P. vranovensis*

Despite worms' naïve attraction to *Pseudomonas vranovensis*, this bacterium is pathogenic to *C. elegans*: adult exposure to *P. vranovensis* ("GRb0427" hereafter) causes severe illness (**Fig 2A**) and significantly reduces survival to less than 2–3 days (**Fig 2B**), in contrast to *C. elegans'* normal lifespan of 2–3 weeks.

Exposure to *P. aeruginosa* PA14 causes gene expression changes in particular *C. elegans* sensory neurons; specifically, a 24-hour exposure to PA14 results in the induction of expression of the TGF-beta ligand DAF-7, as indicated by *daf-7p*::*gfp*, in the ASJ neurons and an increase in *daf-7p*::*gfp* expression in the ASI neurons [27,59]. PA14 small RNAs induce expression of *daf-7p*::*gfp* in the ASI that persists in the F1-F4 progeny generations [23, 27], while the increase in ASJ *daf-7* is caused by PA14 secondary metabolites phenazine-1-carboxamide and pyochelin [59], and does not persist beyond the P0 [27]. To determine whether altered *daf-7* levels correlate with the learned avoidance response to *P. vranovensis*, we examined *daf-7p*::*gfp* expression in GRb0427-trained animals (P0). Upon GRb0427 exposure, *daf-7p*::*gfp* levels significantly increase in the ASI neurons (**Fig 2C and 2D**), but no expression was observed in the ASJ neurons, in contrast to the response to PA14 training [27] (**Fig 2E**). Although *daf-7p*:*GFP* levels increase only in the ASI neurons upon GRb0427 exposure, this increase is comparable to that observed upon PA14 exposure (**Fig 2F**).

Unlike other pathogenic bacteria, exposure to GRb0427 triggers a significantly milder innate immune response, as indicated by low expression of the innate immune response *irg-1* (Infection Response Gene) promoter-GFP reporter, which is induced upon exposure to PA14 but not GRb0427 (**Fig 2G and 2H**). Lack of induction of phenazine-mediated ASJ *daf-7p*::*gfp* expression and only a mild induction of the *irg-1* dependent innate immune pathway suggest that these innate immune pathways might not play a significant role in the neuronal response to the wild bacteria *P. vranovensis* GRb0427, even in the P0 generation, unlike the response to the clinical isolate PA14.

## The avoidance response to *P. vranovensis* is transmitted for four generations

Since training on *P. vranovensis* resulted in an increase in P0 *daf-7p*::*gfp* levels in the ASI (Fig 2C and 2D), and the adult F1 progeny of GRb0427-trained mothers showed robust avoidance of *P. vranovensis* compared to the F1 progeny of the control (OP50-trained) mothers (Fig 1) we examined the expression of *daf-7p*::*gfp* in progeny of GRb0427-trained mothers: these F1 animals express higher levels of *daf-7p*::*gfp* in the ASI neurons (**Fig 3A and 3B**) compared to that in F1 animals from OP50-trained mothers.

To examine if learned avoidance to GRb0427 is inherited transgenerationally (beyond the F1 generation), we first asked whether *daf-7p*::*gfp* expression in the ASI remains high in the

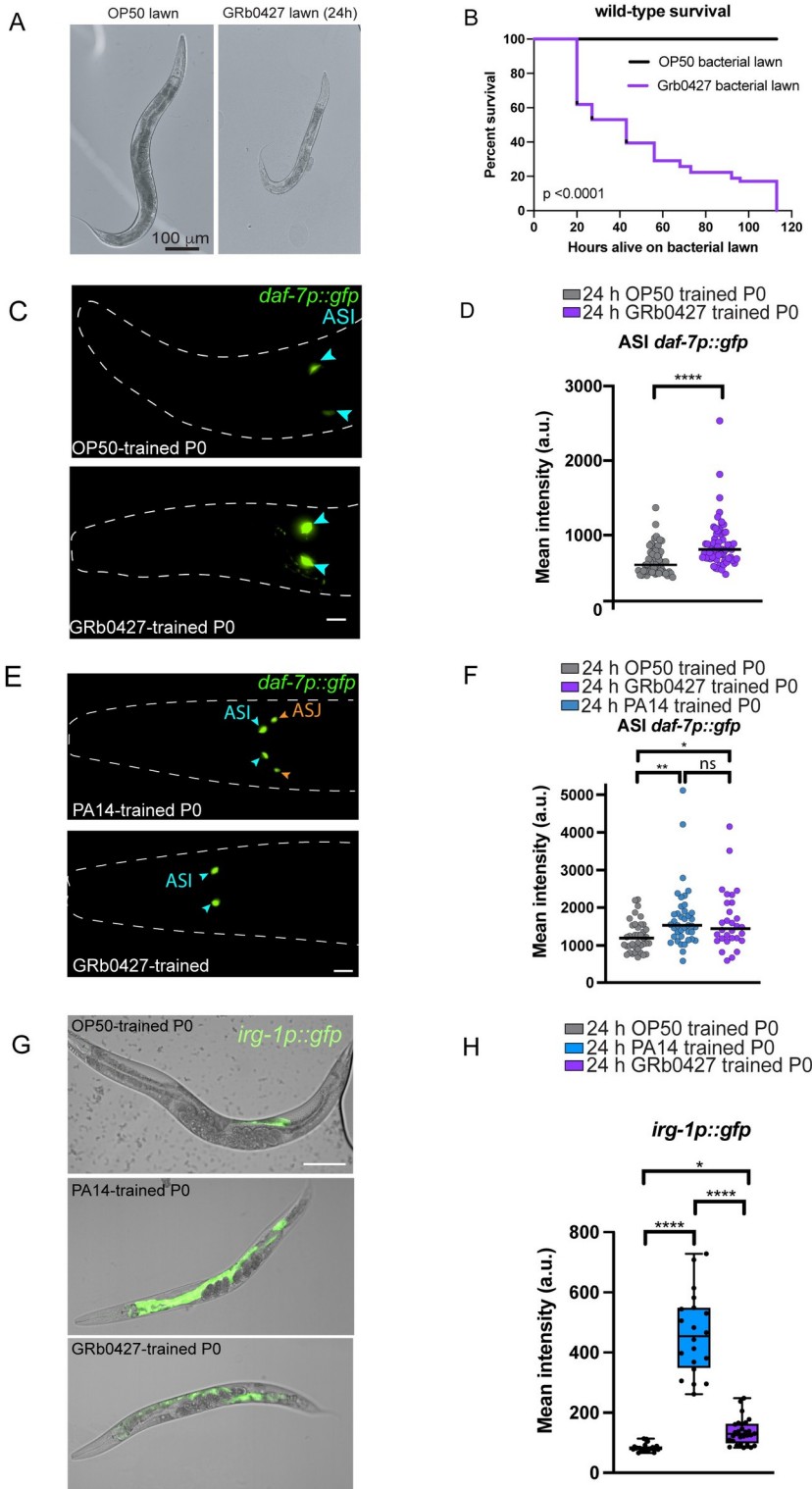

**Fig 2. GRb0427, a natural Pseudomonad pathogen of *C. elegans* induces learned avoidance. (A)** Representative images acquired after exposing Day 1 worms to OP50 (left) or GRb0427 (right) for 24 hours. GRb0427 is pathogenic and 24-hour exposure makes worms sick. **(B)** Worms have significantly lower survival on a GRb0427 lawn compared to an OP50 lawn (p<0.0001 by Log-rank (Mantel-Cox)). **(C)** Representative image of a 24-hour OP50-trained or GRb0427-trained worm expressing *daf-7p::gfp*, which is expressed in the ASI sensory neurons (blue arrowheads). The

dashed line indicates the outline of the worm head. Scale bar = 10 μm. **(D)** Quantification of mean ASI *daf-7p::gfp* intensities from OP50 and GRb0427 animals shows higher expression in GRb0427-trained animals. **(E)** Representative image of a 24-hour PA14-trained (top) or GRb0427-trained (bottom) worm. *daf-7p::GFP* is expressed in the ASI (blue arrowheads) and ASJ (orange arrowheads) sensory neurons (right panel) in the PA14-trained worm, but only in the ASI (blue arrowheads) in the GRb0427-trained worm. The dashed line indicates the outline of the worm head. Scale bar = 10 μm. **(F)** Quantification of mean ASI *daf-7p::gfp* intensities from OP50, PA14, and GRb0427 animals shows similar increase in *daf-7p* expression in PA14 and GRb0427-trained animals. **(G)** Expression of an *irg-1p::gfp* reporter in representative OP50-trained, PA14-trained and GRb0427-trained animals. Images are merged confocal micrographs of brightfield and GFP channels. Scale bar = 100 μm. **(H)** Mean fluorescence intensity of an *irg-1p::gfp* innate immune response reporter in OP50 (gray), *Pseudomonas aeruginosa* PA14 (blue), and GRb0427 (purple)-trained worms. Mean *irg-1p::gfp* reporter intensity is significantly lower in GRb0427-trained worms compared to PA14-trained worms. Each dot represents an individual neuron (D, F) or an individual worm (H). Boxplots: center line, median; box range, 25th–75th percentiles; whiskers denote minimum-maximum values. Unpaired, two-tailed Student's t test, ****p < 0.0001 (D, F); one-way ANOVA with Tukey's multiple comparison's test, ****p<0.0001, **p<0.01, *p < 0.05, ns, not significant (H). For the survival assay in (B), ****p<0.0001 (by Log-rank (Mantel-Cox) test for survival).

grandprogeny and subsequent progeny of GRb0427-trained mothers; like F1, the F2 and F4 animals had higher levels of ASI *daf-7p::gfp*, and these levels return to baseline (similar to the OP50 control) in the F5 generation (**Figs 3C, 3D** and S2**).** We then tested the next generations of progeny for avoidance. The learned avoidance of GRb0427 lasts up to the F4 generation, but returns to naïve attraction to GRb0427 in the F5 generation (**Fig 3E**). Thus, GRb0427 training induces transgenerational inheritance of learned avoidance behavior, as we previously found for PA14. Notably, in contrast to the higher avoidance of PA14 in the P0 generation than in F1-F4 [27], the level of avoidance of *P. vranovensis* is constant across P0 through F4 (**Fig 3F**). This result is consistent with the ASI (but not ASJ) expression of *daf-7p::gfp* and lack of expression of the innate immunity reporter *irg-1p::gfp* (Fig 2G and 2H) suggesting that innate immunity pathways may not contribute significantly to *C. elegans'* avoidance of *P. vranovensis*, but rather that the major pathway of avoidance of *P. vranovensis* even in the first generation is through the same pathway as in F1-F4, rather than through classical innate immune pathways. This lack of an innate immune response to GRb0427 is particularly notable since *P. vranovensis* is found in *C. elegans'* natural habitat [47], while PA14, the standard pathogen used for worm host-pathogen studies, is a human clinical isolate of *P. aeruginosa*.

## *P. vranovensis* avoidance requires the *Cer1* retrotransposon and can be horizontally transferred

We had previously shown that the *Cer1* retrotransposon is required for the learned avoidance of PA14 and its transgenerational inheritance [26] and is proposed to be involved in the transmission of information from the germline to neurons [26]. Similarly, we found that learned avoidance of *P. vranovensis* and the transgenerational inheritance of this avoidance require *Cer1* (**Fig 3G–3I**). We had also found that training worms on conditioned media from the progeny of PA14-trained mothers can induce avoidance [26]; similarly, conditioned media from progeny of GRb0427-trained mothers can also induce learned avoidance (**Fig 3J**), indicating that the learned information can be horizontally transferred.

## Wild *C. elegans* strains can learn to avoid *P. vranovensis* and transgenerationally transmit this information

Wild *C. elegans* strains have been isolated all over the world [60,61], and can be helpful in distinguishing lab N2-strain-specific phenomena from those that are likely to function in the wild. We tested JU1580, a wild strain, for its responses to *P. vranovensis*; we find that JU1580, like N2, is attracted to GRb0427, but learns to avoid it after 24hr of training (**Fig 4A**). Like N2,

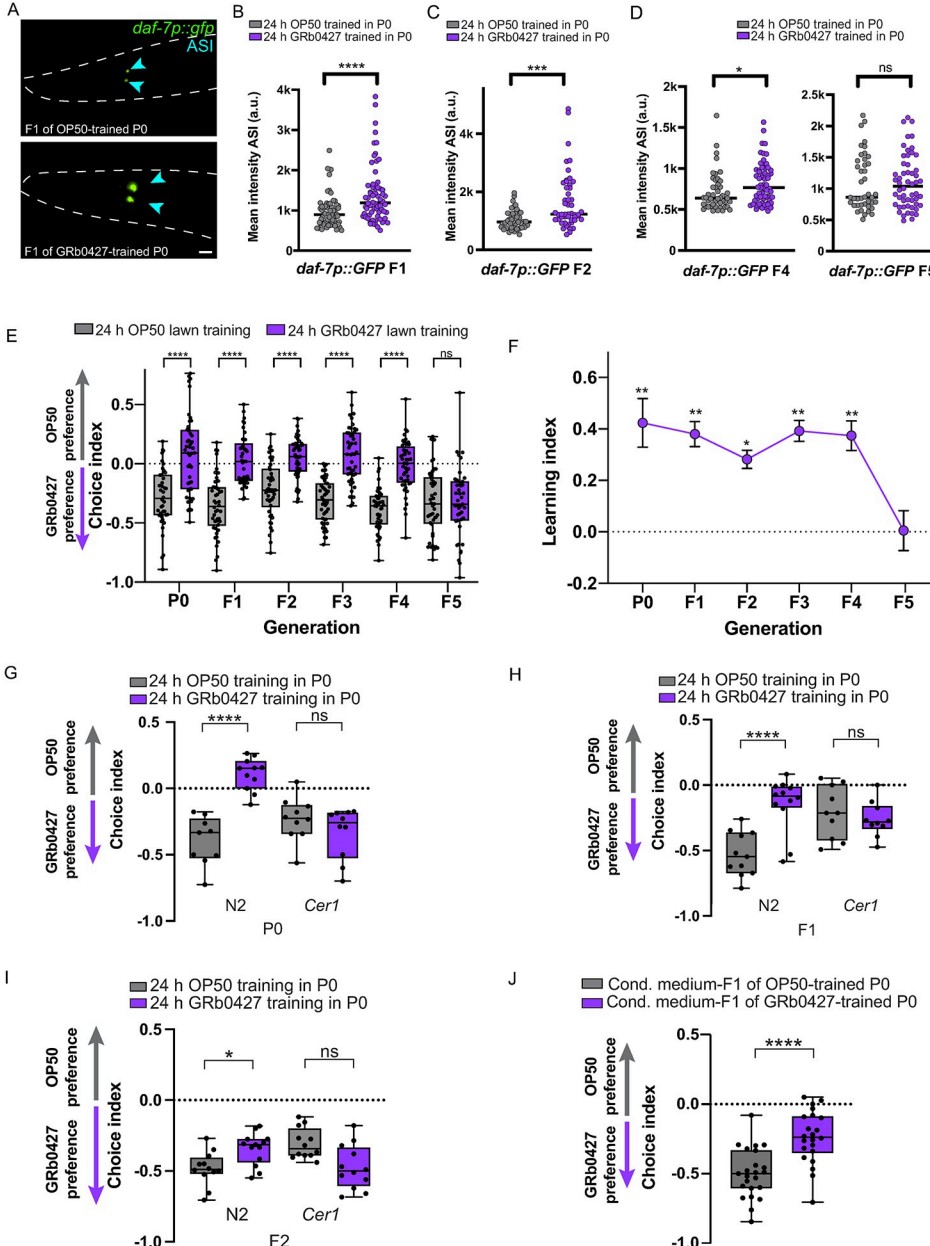

**Fig 3. GRb0427-mediated learned avoidance is inherited transgenerationally. (A)** F1 progeny of GRb0427-trained animals have higher *daf-7p::gfp* expression in the ASI sensory neurons (blue arrowheads). Scale bar = 10 μm. **(B)** Quantification of mean ASI *daf-7p::gfp* intensities from F1 progeny of OP50-trained and GRb0427 animals shows higher expression in F1 progeny of GRb0427-trained animals. **(C, D)** Quantification of mean ASI *daf-7p::gfp* intensities shows higher expression in F2 (C), F4 (D), and F5 (D) progeny of GRb0427-trained mothers, compared to the respective OP50 controls. **(E)** Untrained F1-F4 progeny of GRb0427-trained P0 animals avoid GRb0427 relative to the progeny of OP50-trained control P0 animals. This avoidance is lost in the F5 generation. **(F)** Learning index (trained choice index - naive choice index) of generations P0–F5. Error bars represent mean ± SEM. **(G-I)** N2 (wild-type) worms trained on a GRb0427 bacterial lawn learn to avoid GRb0427, but *Cer1(gk870313)* mutant animals don't exhibit learned avoidance (G). The learned avoidance is inherited by the F1 (H) and F2 progeny (I) of GRb0427-trained N2 mothers but not by the progeny of GRb0427-trained *Cer1(gk870313)* mothers (n = 1). **(J)** Worms exposed to conditioned medium from the F1 progeny of OP50- and GRb0427-trained mothers were tested in a choice assay between OP50 and GRb0427. Worms exposed to conditioned medium from the F1 progeny of GRb0427-trained mothers exhibit avoidance of GRb0427. Each dot represents an individual choice assay plate (E, G-J) or an individual neuron for fluorescence images (B-D). Boxplots: center line, median; box range, 25th–75th percentiles; whiskers denote minimum-maximum values. Unpaired, two-tailed Student's t test, ****p < 0.0001, ***p<0.001, *p<0.05, ns, not significant (B-D, J); one-way ANOVA with Tukey's multiple comparison's test, ****p<0.0001, **p<0.01, *p<0.05,

ns, not significant (E, F), Two-way ANOVA with Tukey's multiple comparison's test, \*p<0.05, \*\*\*\*p<0.0001, ns, not significant (G-I).

JU1580 worms inherit this learned avoidance through the F4 generation, then return to naïve attraction in the F5 (**Fig 4B**). We tested an additional wild strain, ED3040 [49], and found that it behaved similarly to N2 and JU1580 in its initial attraction to *P. vranovensis* and its learned avoidance after 24hr of exposure (**Fig 4C**). Thus, it is likely that many wild *C. elegans* strains are attracted to *P. vranovensis*, learn to avoid it after exposure, and can transmit this learned avoidance transgenerationally, as we have shown for the laboratory *C. elegans* strain, N2.

## *P. vranovensis* small RNAs drive learned avoidance

Since learned avoidance to *P. vranovensis* is transgenerationally inherited, and transgenerational inheritance of avoidance of PA14 is driven by its small RNA, P11, we next asked if small RNAs made by *P. vranovensis* induce avoidance. Like PA14, when adult *C. elegans* were exposed for 24 hours to sRNAs isolated from *P. vranovensis* GRb0427, worms learned to avoid *P. vranovensis* (**Fig 5A**). Exposure to *P. vranovensis* sRNAs increased *daf-7p::gfp* expression in the ASI sensory neurons (**Fig 5B and 5C**). We next tested if *P. vranovensis* sRNA-induced learned avoidance is transgenerationally inherited. Indeed, as observed for *P. vranovensis* lawn exposure, *P. vranovensis* sRNA-induced avoidance is inherited up to the F4 generation and resets in the F5 (**Fig 5D and 5E**). Training on GRb0427 small RNAs also induces learned avoidance and transgenerational inheritance in JU1580 worms (**Fig 5F and 5G**).

## *P. vranovensis* sRNA treatment also induces avoidance of PA14

We next asked if learned avoidance induced by *P. vranovensis* small RNAs is species-specific. We trained worms on *P. vranovensis* sRNAs and tested avoidance of

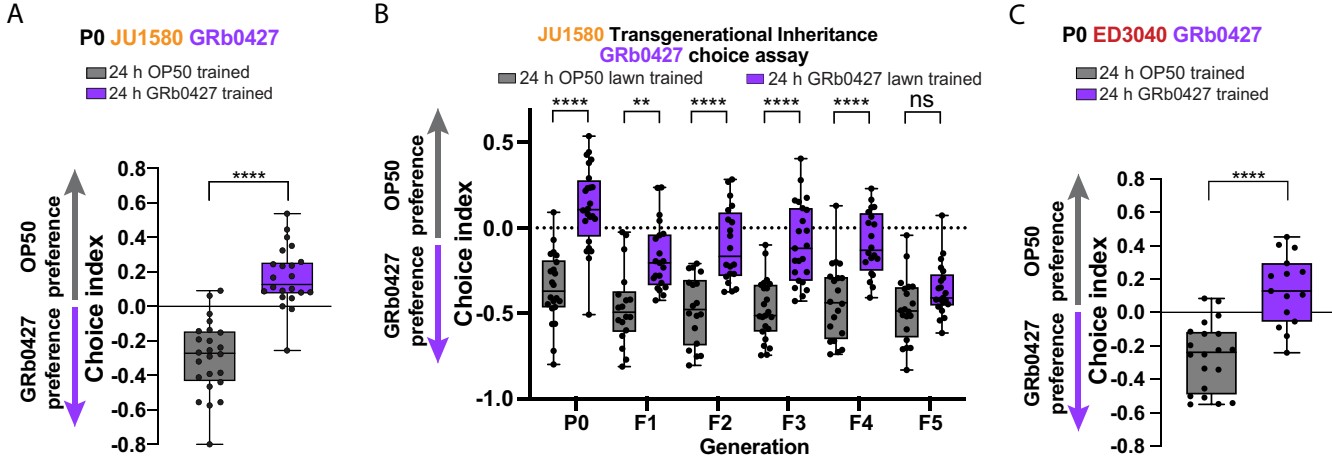

**Fig 4. Wild *C. elegans* strains can learn to avoid *P. vranovensis* and transgenerationally transmit this information.** (**A**) A wild strain of *C. elegans*, JU1580 (a natural *C. elegans* isolate), avoids GRb0427 following 24-hour exposure to a GRb0427 bacterial lawn. (**B**) Untrained F1-F4 progeny of GRb0427 bacterial lawn-trained P0 JU1580 animals avoid GRb0427 relative to the progeny of OP50 bacterial lawn-trained control P0 JU1580 animals. This avoidance is lost in the F5 generation. (**C**) ED3040 (another natural isolate of *C. elegans*) also learns to avoid GRb0427 after a 24-hour exposure to a GRb0427 lawn. Each dot represents an individual choice assay plate. Boxplots: center line, median; box range, 25th–75th percentiles; whiskers denote minimum-maximum values. Unpaired, two-tailed Student's t-test (A, C), \*\*\*\*p<0.0001; one-way ANOVA with Tukey's multiple comparison's test, \*\*\*\*p<0.0001, \*\*p<0.01, ns, not significant (B).

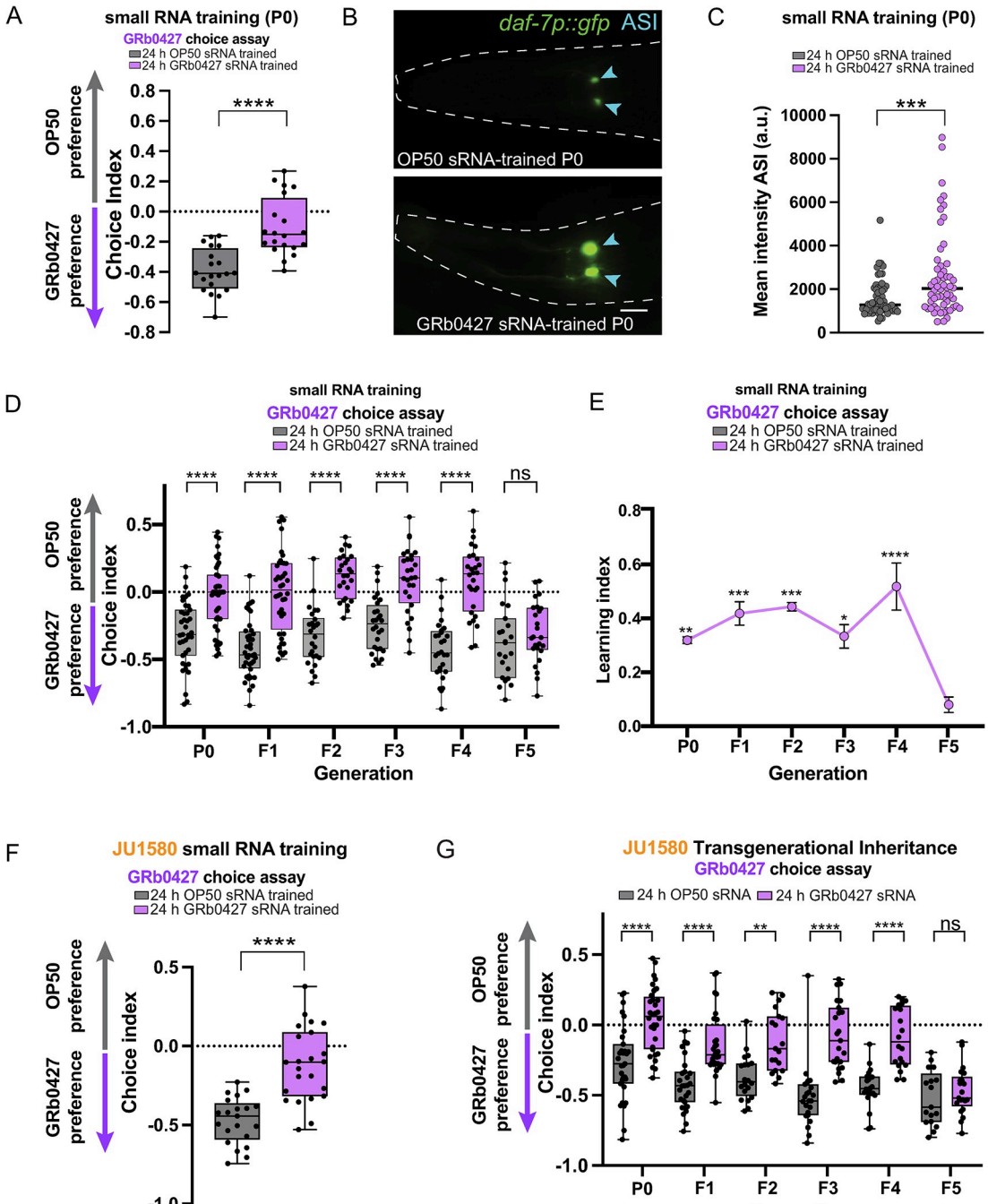

**Fig 5. GRb0427 small RNAs induce learned avoidance and its transgenerational inheritance.** (**A**) Worms trained on GRb0427 small RNAs exhibit learned avoidance of GRb0427 in an OP50-GRb0427 choice assay. (**B**) GRb0427 sRNA-trained animals have higher *daf-7p::gfp* expression in the ASI sensory neurons (blue arrowheads). Scale bar = 10 μm. (**C**) Quantification of mean ASI *daf-7p::gfp* intensities from OP50 sRNA-trained and GRb0427 sRNA-trained animals shows higher expression in GRb0427 sRNA-trained animals. (**D**) Untrained F1-F4 progeny of GRb0427 sRNA-trained P0 animals avoid GRb0427 relative to the progeny of OP50 sRNA-trained control P0 animals. This avoidance is lost in the F5 generation. (**E**) Learning index (index - naive choice index) of generations P0–F5. Error bars represent mean ± SEM. (**F**) GRb0427 sRNA-trained JU1580 animals avoid GRb0427 relative to OP50 sRNA-trained JU1580 animals. (**G**) F1-F4 progeny of GRb0427 sRNA-trained JU1580 animals avoid GRb0427 relative to the respective controls. This avoidance is lost in the F5 generation. Each dot represents an individual choice assay plate (A, D, F, G) or an individual neuron for fluorescence images (C). Boxplots: center line, median; box range, 25th–75th percentiles; whiskers denote minimum-maximum values. Unpaired, two-tailed Student's t test (A, C, F), ***p<0.001, ****p<0.0001; one-way ANOVA with Tukey's multiple comparison's test, ****p< 0.0001, ***p<0.001, **p<0.01, *p<0.05, ns, not significant (D, E, G).

PA14; *P. vranovensis* sRNA training induces avoidance to PA14, similar to training on PA14 sRNAs (**Fig 6A**). Previously, we found that a specific PA14 small RNA, P11, with 17nt of perfect match to the *C. elegans* neuronal homolog of the human nervous-system-specific ER membrane protein Macoilin, *maco-1* [62], is required for learned avoidance [23]. Intriguingly, worms exposed to bacteria expressing only the PA14 sRNA P11 also induces avoidance to *P. vranovensis* (**Fig 6B**), suggesting that the underlying mechanism in *P. vranovensis* sRNA-induced avoidance is like that of PA14-induced avoidance.

## *P. vranovensis* treatment decreases *maco-1* transcripts through F4

Consistent with the idea of a conserved mechanism between PA14 and *P. vranovensis*-induced avoidance, loss of *maco-1*, either by mutation *(maco-1(ok3165))* (**Fig 6C**) or by RNAi treatment (**Fig 6D**) significantly reduces the strong naïve preference for GRb0427. Similarly, exposure to *P. vranovensis* small RNAs does not further increase *maco-1* mutants' avoidance of *P. vranovensis* (**Fig 6C**), indicating that *maco-1* loss phenocopies GRb0427 sRNA treatment. Quantitative RT-PCR showed a decrease in relative *maco-1* transcript abundance in GRb0427-trained animals (**Fig 6E**), as well as in the F2 and F4 progeny of GRb0427-trained mothers (**Fig 6F**), but levels of *maco-1* return to the same levels as untrained animals in the F5 generation (**Fig 6F**). That is, the decrease in *maco-1* transcript levels after P0 treatment on *P. vranovensis* persists from F0 through F4 generation, mirroring the change in avoidance. This observation indicates that learned avoidance induced by *P. vranovensis* targets *maco-1*, as PA14's P11 small RNA does. We also examined differentially-expressed genes between *P. vranovensis*-treated and *E. coli* HB101-treated P0 adult worms in RNA sequencing data reported in Burton et al., 2020 [8]; consistent with our results, *maco-1* expression is downregulated in *P. vranovensis*-treated adult animals ($log_2$fold change = -0.2837998, padj = 0.027) in this independent analysis [8]. (Our previous experiments suggest that we should only see changes in adult animals with fully-developed germlines [27]; the Burton dataset only provided adult data for the P0 generation [7,8])

We next examined whether *P. vranovensis* might encode a P11-like small RNA. We first analyzed the recently-sequenced genome of *P. vranovensis* [8], but we did not find any genomic region with sequence homology to P11. In fact, there is no region analogous to the operon that contains P11 in the *P. vranovensis* genome. That is, while P11 can induce similar avoidance of GRb0427 as it does for PA14, and GRb0427 induces avoidance through a small RNA, GRb0427 does not appear to encode a small RNA with a P11-like sequence in its genome; therefore, we needed to determine whether GRb0427 expresses a different sRNA that induces learned avoidance and inheritance of this avoidance.

While a P11-like sRNA cannot account for the learned avoidance of GRb0427, our small RNA *maco-1* experiments suggested that an sRNA with similarity to *maco-1* might be involved. Therefore, we searched the *P. vranovensis* genome for similarity to the *maco-1* sequence; we found five perfect matches to the *maco-1* coding region in the *P. vranovensis* genome, but only one of these, a 16nt match, lies in an intergenic region that would be likely to encode a small RNA (**Fig 7A–7C**). Interestingly, this sequence identity lies in a different exon of *maco-1* (Exon 1) from P11's 17nt perfect match (Exon 8; **Fig 7A**).

The *P. vranovensis* intergenic region containing this 16-nucleotide sequence match to *maco-1* is flanked by bacterial protein-coding genes with predicted functions in the iron metabolism and sugar transport pathways (**Fig 7B**). To test whether the region containing this match might mediate *P. vranovensis*-induced avoidance, we expressed a 347 bp region of this intergenic sequence ("IntReg"; **Fig 7C**) in *E. coli*, and found that training on IntReg induces avoidance to *P. vranovensis* (**Fig 7D**). The avoidance induced by *E. coli* expressing the

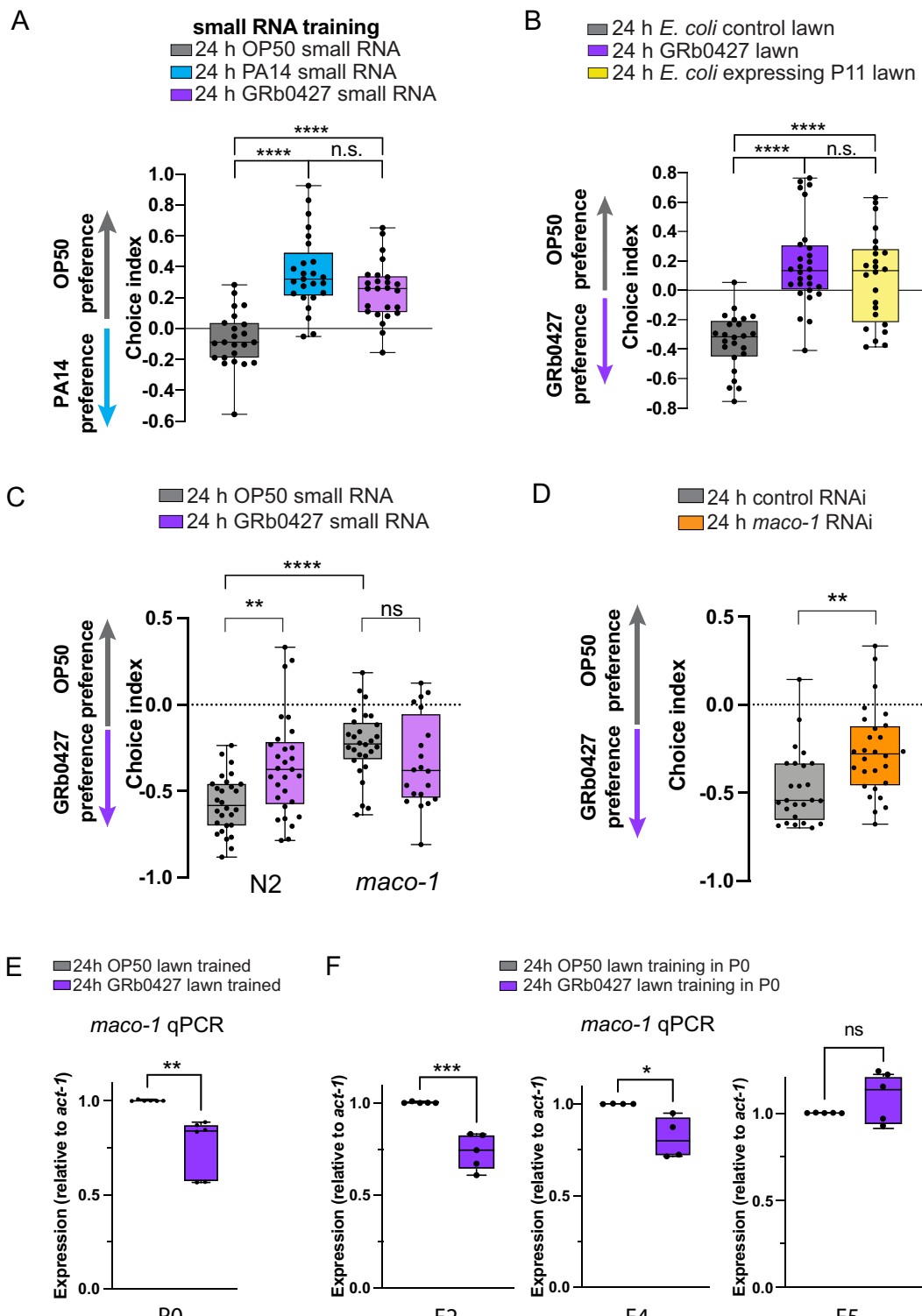

**Fig 6. GRb0427 sRNA-induced avoidance requires *maco-1* and GRb0427 training heritably reduces *maco-1* transcripts.**
**(A)** Worms were trained on OP50 (gray), *Pseudomonas aeruginosa* PA14 (blue), or GRb0427 (purple) small RNAs, and tested for their PA14 preference in a bacterial choice assay between OP50 and PA14. Worms treated on GRb0427 small RNAs not only avoid GRb0427 (Fig 3A), but also avoid PA14 in an OP50-PA14 choice assay. **(B)** Worms trained on the PA14 small RNA P11 avoid GRb0427 in an OP50-GRb0427 choice assay. **(C)** N2 (wild-type) worms trained on GRb0427 small RNAs avoid

GRb0427, relative to Worms trained on OP50 small RNAs. *maco-1(ok3165)* loss-of-function mutant worms naively avoid GRb0427, relative to wild type worms, and do not show increased avoidance upon exposure to GRb0427 sRNAs. **(D)** Upon downregulation of *maco-1* by RNAi, wild type worms exhibit higher naïve avoidance of GRb0427 compared to control RNAi-treated wild-type worms. **(E)** Fold change $(2^{-\Delta\Delta C_t})$ of *maco-1* transcript levels in GRb0472-trained P0 (E), F2, F4, and F5 animals (F) relative to the respective OP50-trained controls (*act-1* was used as the housekeeping gene for reference). Each data point represents an independent biological replicate, and 3 technical replicates were performed for each biological replicate. Each dot represents an individual choice assay plate (A-D) or a biological replicate in a qPCR assay (E,F). Boxplots: center line, median; box range, 25th–75th percentiles; whiskers denote minimum-maximum values, One-way ANOVA with Tukey's multiple comparison's test, ****$p<0.0001$, ns, not significant (A,B); Two-way ANOVA with Tukey's multiple comparison's test, **$p<0.01$, ****$p<0.0001$, ns, not significant (C); Unpaired, two-tailed Student's t test, ***$p<0.001$, **$p<0.01$, *$p<0.05$ (D-F).

intergenic region persists for four generations after parental exposure and is lost by the F5 generation (**Fig 7E**), similar to the transgenerational inheritance of learned avoidance induced by a *P. vranovensis* lawn or small RNA exposure. Like *P. vranovensis* treatment, exposure of the wild strain JU1580 to the IntReg clone also induced avoidance of P. vranovensis (**Fig 7F**).

## A specific *P. vranovensis* sRNA induces learned avoidance

We next investigated if the intergenic region encodes a small RNA. While the genome of *P. vranovensis* is published, no information on small RNAs were publicly available, so we sequenced the small RNAs produced by *P. vranovensis* (see Methods), that is, the total small RNA pool that induces learned avoidance and its transgenerational inheritance. Indeed, we detected a small RNA that maps within the intergenic region that contains the 16-nucleotide sequence match to *maco-1* (**Fig 8A**). This small RNA, which we named "Pv1", is 124 bp long, and its homology to *maco-1*, like P11, lies in a predicted stem loop (**Fig 8B**).

We next asked whether exposure to Pv1 expressed in *E. coli* would be sufficient to induce avoidance to *P. vranovensis;* indeed, training on *E. coli-Pv1* induces avoidance in the mother generation (P0; **Fig 8C**), as well as the inter (F1)- and transgenerational (F2) inheritance of this learned avoidance (**Fig 8D**).

To determine if Pv1's sequence identity to *maco-1* is necessary for the learned avoidance to *P. vranovensis* and its transgenerational inheritance, we constructed a mutant *P. vranovensis* bacterial strain lacking the 16-nucleotide match to *maco-1*, Δ16. This Δ16 mutant GRb0427 strain is equally attractive to worms as wild-type GRb0427, and *C. elegans* prefer Δ16 to OP50 just as they prefer GRb0427 to OP50 (**Fig 8E–8G**), suggesting that the bacteria do not "smell" different to the worms. Additionally, Δ16 is similarly pathogenic to worms as wild-type GRb0427 (**Fig 8H**). However, Δ16 is unable to induce transgenerational inheritance of learned avoidance (**Fig 8I**), nor can Δ16 induce *daf-7p::gfp* expression in ASI neurons (**Fig 8J**).

Finally, mismatch of four of the nucleotides in the loop of Pv1 within the 16nt *maco-1* match (but that still retains its predicted stem-loop structure, **Fig 8K**) removes the ability of Pv1 to induce avoidance (**Fig 8L**). Together, our data suggest that a specific *P. vranovensis* small RNA, Pv1, is sufficient to induce avoidance, and Pv1's 16nt match to the *maco-1* sequence is necessary for sRNA-mediated learned avoidance of GRb0427 and its transgenerational inheritance.

## Mechanism: RNAi components are required for Pv1-induced avoidance

Previously, we showed that components of the RNA interference pathway are required for PA14- and P11-induced learned avoidance and its transgenerational inheritance [23,27]. These components included the SID-2 dsRNA transporter, the DCR-1 (Dicer) endoribonuclease, and the SID-1 RNA transmembrane dsRNA transporter. To determine whether the *P. vranovensis* sRNA Pv1 is processed through a similar mechanism, we tested mutants of these

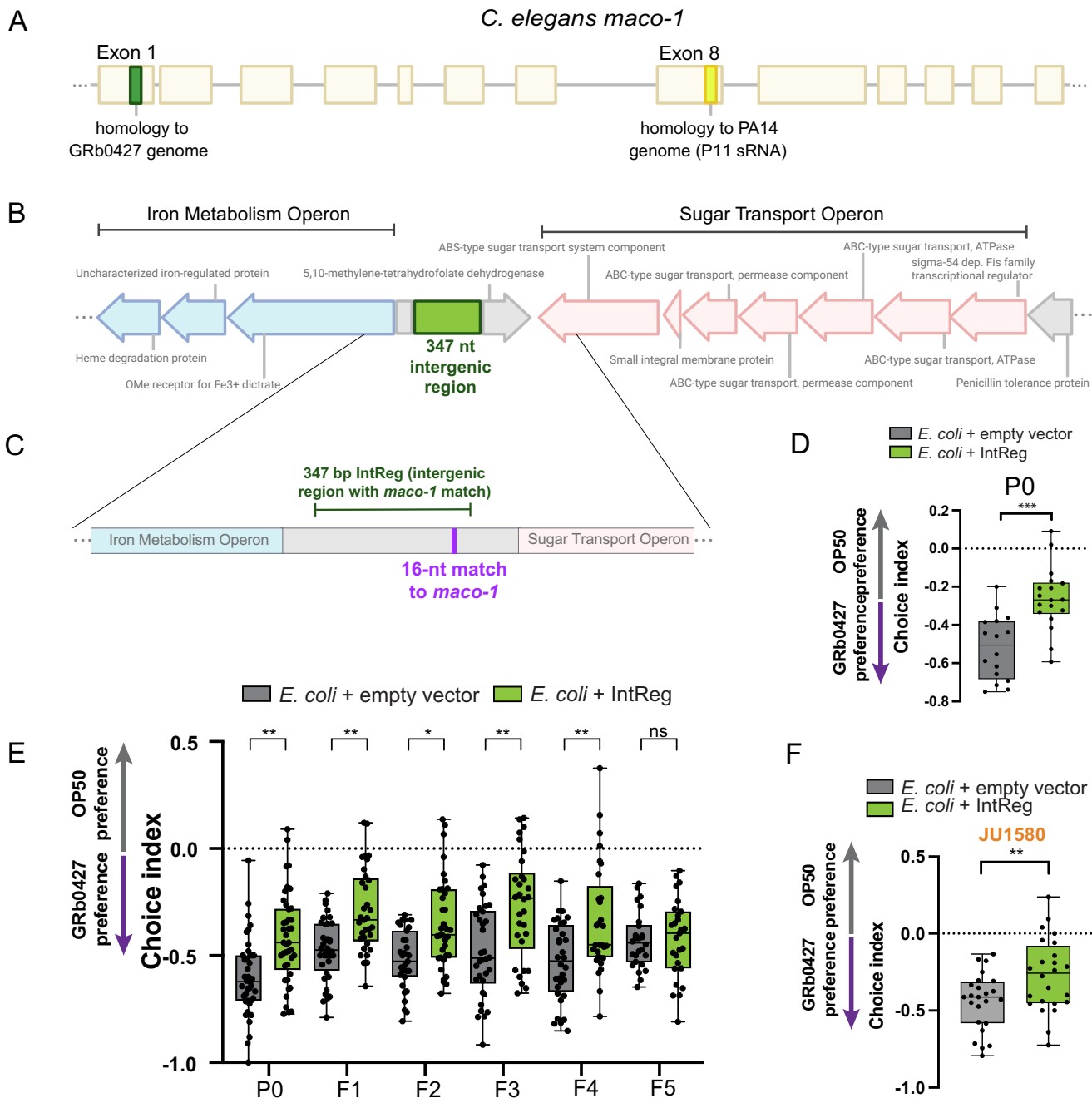

**Fig 7. An intergenic region in the GRb0427 genome contains a 16-nucleotide perfect match to *maco-1* and is sufficient for learned avoidance of GRb0427.** **(A)** The PA14 small RNA P11 contains a 17-nucleotide perfect match to the worm neuronal gene *maco-1* (Exon 8). We found an intergenic region in the GRb0427 genome with a 16-nucleotide perfect match to a stretch of Exon 1 of *maco-1*. **(B, C)** The GRb0427 genome has an intergenic region, flanked by iron metabolism operon and sugar transport operons, containing a 16-nucleotide perfect match to *maco-1* (B). This region is represented as a schematic in (C). 347 bp of this intergenic region ("IntReg", shown in green) was cloned into *E. coli* for testing. IntReg contains the 16-nt match to *maco-1* (indicated in purple). **(D)** Training worms on *E. coli* expressing the intergenic region (IntReg) with the match to *maco-1* induces avoidance of GRb0427. **(E)** Untrained F1-F4 progeny of worms trained on *E. coli* expressing the intergenic region with the match to *maco-1* exhibit higher avoidance of GRb0427 compared to controls. This higher avoidance is lost in the F5 generation. **(F)** Training of JU1580 worms on *E. coli* expressing the intergenic region (with the match to *maco-1*) induces avoidance of GRb0427. Each dot represents an individual choice assay plate (D-F). Box plots: center line, median; box range, 25th–75th percentiles; whiskers denote minimum-maximum values. Unpaired, two-tailed Student's t test, **$p<0.01$, ***$p<0.001$ (D, F); one-way ANOVA with Tukey's multiple comparisons test, **$p<0.01$, *$p<0.05$ ns, not significant (E).

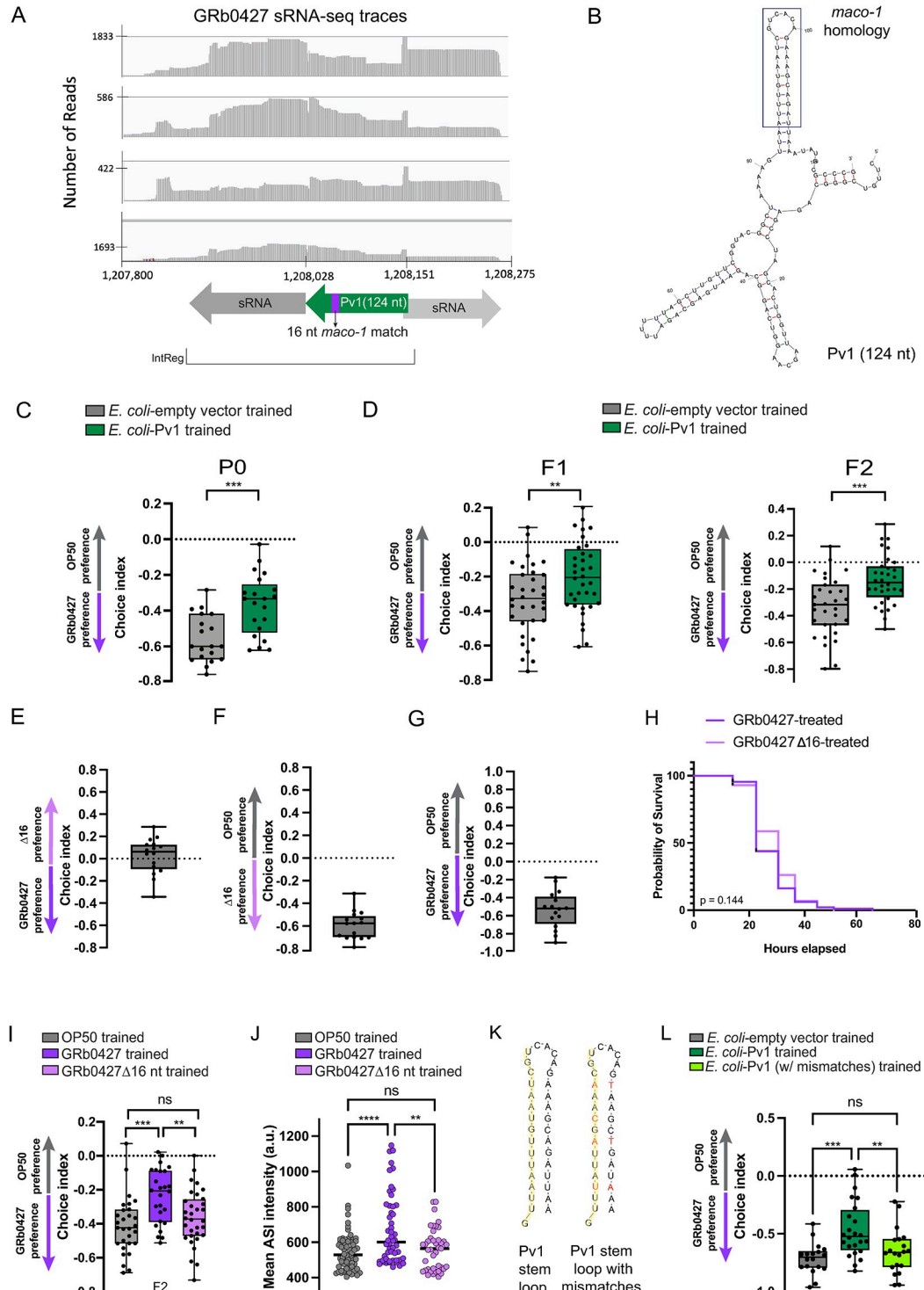

**Fig 8. Pv1, a GRb0427 sRNA, is necessary and sufficient for transgenerational inheritance of learned avoidance. (A)** Alignment of GRb0427 small RNA sequencing reads to the GRb0427 genome. The peaks obtained from this alignment and the Sapphire small RNA promoter prediction software indicated the presence of three small RNAs (shown in dark gray, green, and light gray) in the intergenic region between the two predicted operons described in Fig 7. Of the 3 predicted sRNAs, the sRNA marked in green contains the 16-nucleotide match to *maco-1* (purple), and we named it Pv1. **(B)** mFold structure prediction for Pv1; the *maco-1* region is in a predicted stem loop of Pv1 (the boxed region). **(C)** Training worms on *E. coli* expressing just the Pv1 small RNA induces avoidance of GRb0427 (compared to training on *E. coli* expressing a control empty vector). **(D)** *E. coli*-Pv1 induced learned avoidance of GRb0427 is transgenerationally inherited. Untrained

F1 and F2 progeny of *E. coli*-Pv1 trained P0 worms also exhibit higher GRb0427 avoidance compared to controls. **(E-G)** Naïve worms do not exhibit a preference towards either wild type GRb0427 or GRbΔ16nt strain in a GRb0427-GRbΔ16nt choice assay (E), while preferring both GRb0427 and GRbΔ16nt with respect to OP50 (F, G). **(H)** Survival of worms on lawns of GRb0427 and GRbΔ16nt are not significantly different. **(I)** F2 progeny of GRbΔ16nt-trained worms do not exhibit learned avoidance of GRb0427. **(J)** Mean ASI *daf-7p::GFP* intensities in GRbΔ16nt-trained worms are comparable to that of OP50-trained worms, in contrast to worms trained on GRb0427 where the mean ASI *daf-7p::GFP* intensities are higher than that of OP50-trained worms. **(K)** An *E. coli* strain expressing Pv1 containing 7 mismatches (4 of which lie within the 16- nucleotide match to *maco-1*) is predicted to have identical secondary structure to wild type Pv1, but the mismatches disrupt the sequence homology to *maco-1*. **(L)** Worms trained on *E. coli*-Pv1 learn to avoid GRb0427, while worms trained on *E. coli* expressing the Pv1 with mismatches fail to learn avoidance. Each dot represents an individual choice assay plate (C–G, I, L), or an individual neuron (J). Boxplots: center line, median; box range, 25th–75th percentiles; whiskers denote minimum-maximum values. Unpaired, two-tailed Student's t test, **p<0.01, ***p<0.001, (C, D); One-way ANOVA with Tukey's multiple comparison's test, **p<0.01, ***p<0.001, ****p<0.0001, ns, not significant (I, J, L). For the survival assay in (H), ns–not significant (by Log-rank (Mantel-Cox) test for survival).

components. We found that *sid-2(qt42)* [63,64], *dcr-1(mg375)* [65,66], or *sid-1(qt9)* [67] animals were unable to learn *P. vranovensis* avoidance after Pv1 training (**Fig 9A–9C**). Moreover, mutants of *hrde-1(tm1200)*, the nuclear Argonaute that binds 22G RNAs, cannot learn to avoid *P. vranovensis* after training on Pv1 (**Fig 9D**), indicating that HRDE-1 is also required. Together, our data suggest that the mechanism that we had previously identified for PA14-P11 sRNA processing is shared with *P. vranovensis*-Pv1 processing, and involves bacterial sRNA uptake (*sid-2*), dsRNA processing (*dcr-1*), 22G sRNA binding (*hrde-1*), dsRNA transport (*sid-1*), *maco-1* transcript reduction, *Cer1*-mediated germline-to-neuron communication, *daf-7* expression increase in the ASI, and then behavioral switching from attraction to avoidance (**Fig 9E**).

## *C. elegans* avoid the beneficial bacterium *Pseudomonas mendocina* following exposure to GRb0427 or Pv1

In the wild, worms experience a range of temperatures, and bacterial species can be differentially pathogenic at different temperatures. In the case of *P. aeruginosa* PA14, pathogenicity decreases at lower temperatures [23,68], and the PA14 small RNA, P11 is not expressed at lower temperatures [23]. We wondered if this was also the case for *P. vranovensis*/GRb0427 and Pv1, or if other factors, including other members of the microbiome, might influence the cessation of avoidance.

We first asked if temperature affects the pathogenicity of GRb0427, as it does for PA14. Although less pathogenic to worms than 25°C-grown GRb0427, 15°C-grown GRb0427 still kills all the worms in the population before the control (OP50)-grown worms have started dying (**Fig 10A**). Consistent with these results, the Pv1 small RNA is expressed in total RNA pools from both 25°C GRb0427 and 15°C GRb0427 (**S3 Fig**), unlike the differential expression of PA14's P11 sRNA at different temperatures [23]. Consistent with Pv1 being expressed under both temperature conditions, worm populations trained on 25°C and 15°C GRb0427 small RNAs both avoid GRb0427 compared to worms trained on control (OP50) small RNAs (**Fig 10B**). Thus, it seems unlikely that temperature-dependent changes and Pv1 sRNA expression in GRb0427 pathogenicity drive the loss of memory of learned avoidance.

We next asked if GRb0427-induced learned avoidance might alter worms' responses to other bacteria. Several bacterial species in the worm microbiome are beneficial; that is, they increase worm growth rates and progeny production and extend lifespan [47]. In fact, all the wild bacteria we tested are more attractive than OP50 (**Fig 1**). This preference likely evolved as *Pseudomonas* bacteria are sources of nutrition to *C. elegans* in the wild [47]. Therefore, we asked if exposure to the pathogenic *P. vranovensis*/GRb0427 might cause worms to also avoid beneficial bacteria in their microbiome. Some of the beneficial bacterial species in the worm

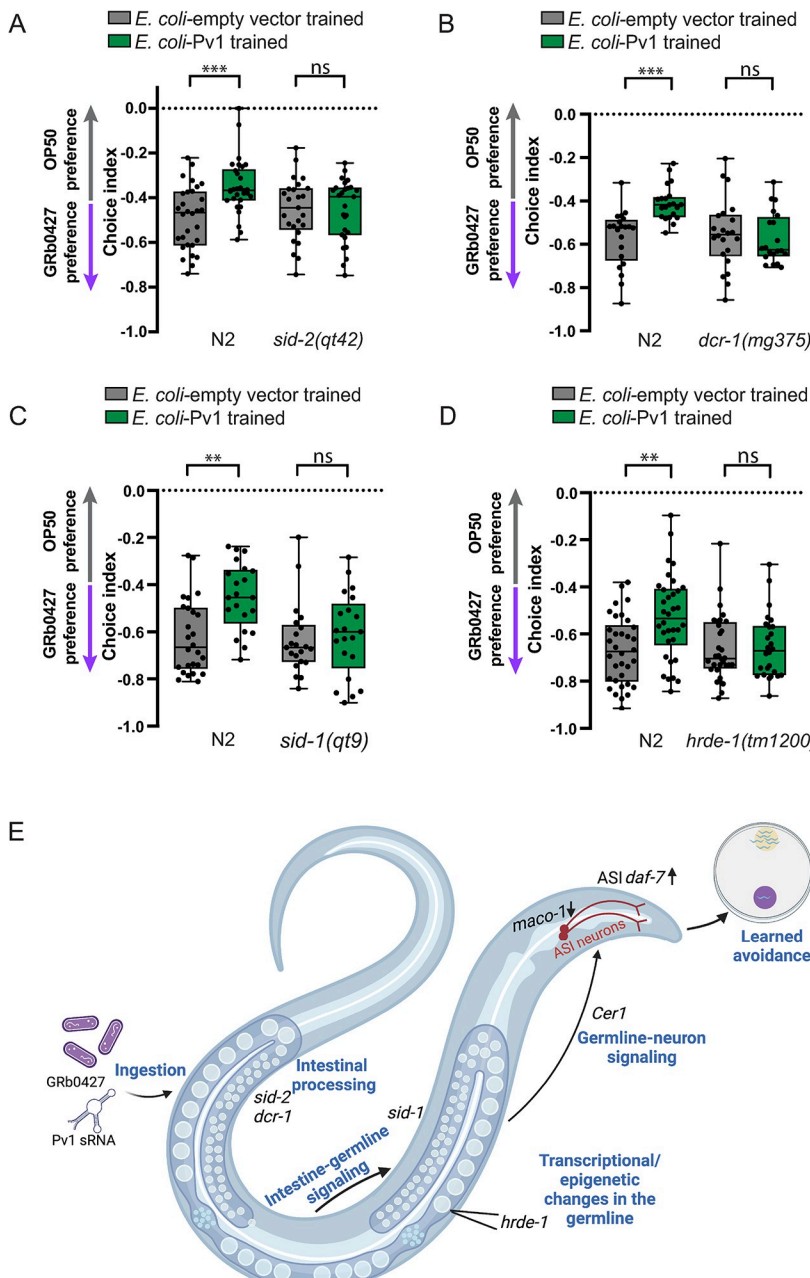

**Fig 9. Genes involved in sRNA processing are required for Pv1 sRNA-mediated learned avoidance. (A-D)** *sid-2 (qt42)* (A), *dcr-1(mg375)* (B), *sid-1(qt9)* (C), and *hrde-1(tm1200)* (D) do not learn to avoid GRb0427 in response to Pv1 training. **(E)** Model of Pv1 uptake, processing, and induced avoidance of *P. vranovensis* (created using Biorender). Boxplots: center line, median; box range, 25th–75th percentiles; whiskers denote minimum-maximum values. Two-way ANOVA with Tukey's multiple comparison's test, **p<0.01, ***p<0.001, ns, not significant (A-D).

microbiome, e.g., *Pseudomonas mendocina*, belong to the same genus as GRb0427 [50]; *C. elegans* are attracted to *Pseudomonas mendocina* [56], and prefer it to OP50 (Figs 1, S4). As we showed above, worms trained on a strain of *P. mendocina* from the worm's microbiome, MSPm1, do not exhibit an altered response to *P. mendocina* compared to control (*E. coli* OP50)-trained worms (**Fig 1B**), as might be expected for beneficial bacteria. Consistent with

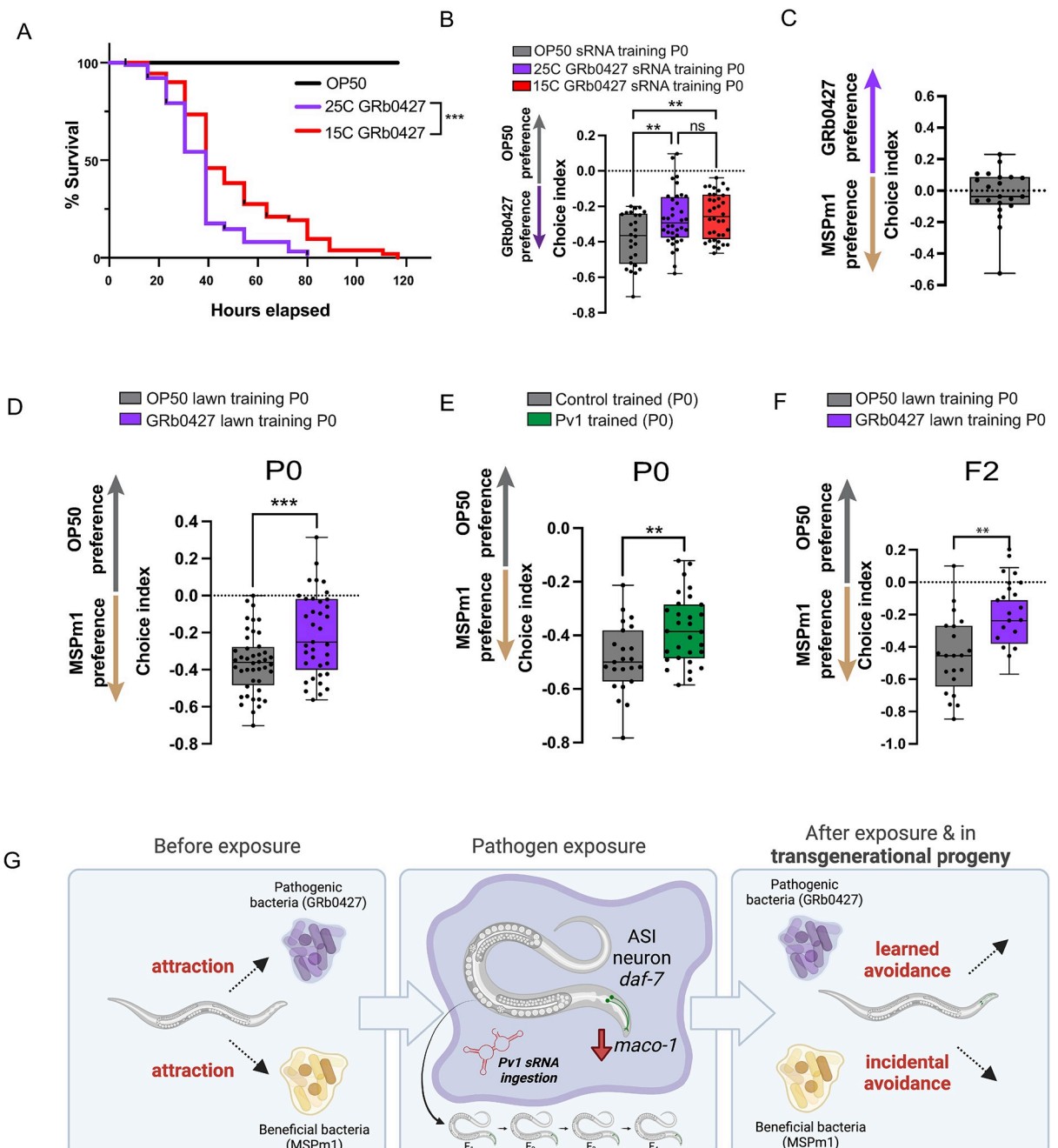

**Fig 10. Pv1-induced learned avoidance causes avoidance of *P. mendocina*, a beneficial natural bacterial species. (A)** Worms have significantly lower survival on a 25˚C-grown GRb0427 lawn compared to a 15˚C-grown GRb0427 lawn, but in both these conditions, all worms die before worms on an OP50 lawn have started dying, indicating that both 25˚C and 15˚C-grown GRb0427 are pathogenic. **(B)** Training worms on sRNAs from both 25˚C and 15˚C-grown GRb0427 induce learned avoidance. **(C)** Untrained worms have similar preference for GRb0427 and MSPm1 in a choice assay. **(D)** Training worms on a GRb0427 lawn reduces attraction to *P. mendocina* compared to OP50-trained control worms. **(E)** Training worms on *E. coli* expressing Pv1 reduces attraction to *P. mendocina*. **(F)** Naïve F2 progeny of GRb0427-trained P0 animals exhibit reduced attraction to *P. mendocina* relative to F2 progeny of OP50-trained P0 animals. **(G)** Schematic outlining the findings of the study. Prior to exposure to pathogenic *P. vranovensis*, worms are naively attracted to both *P. vranovensis*, and the non-pathogenic *P. mendocina*. Exposure to *P. vranovensis* or to the Pv1 small RNA of *P. vranovensis* induces learned avoidance of *P. vranovensis* in worms and four generations of progeny. Exposed worms not only avoid the pathogenic *P. vranovensis*, but also the non-pathogenic *P. mendocina*. Schematic was created using Biorender. Each dot represents an individual choice assay plate (B-F). Boxplots: center line, median; box range, 25th–75th percentiles; whiskers denote minimum-maximum values. One-way ANOVA with Tukey's multiple comparison's test, **p<0.01, ns, not significant (B); Unpaired, two-tailed Student's t test, **p<0.01, ***p<0.001 (D-F). For the survival assay in (A), ***p<0.001 (by Log-rank (Mantel-Cox) test for survival).

this logic, we found that the *P. mendocina* genome does not encode the Pv1 small RNA used by GRb0427 to elicit avoidance, and the intergenic regions in the genome do not contain any significant contiguous matches to the Pv1 target *maco-1*.

We next tested the effect of GRb0427 exposure on worms' responses to *P. mendocina*. In their naïve state, *C. elegans* exhibits no preference for GRb0427 or MSPm1 over one another (**Fig 10C**). However, worms trained on *P. vranovensis*/GRb0427 for 24 hrs have significantly reduced attraction to *P. mendocina* compared to worms grown on the control *E. coli* OP50 (**Fig 10D**)–that is, although they have never been exposed to *P. mendocina*, and it is a beneficial food source they are naively attracted to (Figs 1B and S4), exposure to pathogenic *P. vranovensis* induces avoidance of the beneficial bacteria. Similarly, training on *E. coli* expressing the Pv1 small RNA is sufficient to reduce attraction of worms to the beneficial *P. mendocina* (**Fig 10E**). Since Pv1 mediates transgenerational inheritance of learned GRb0427 avoidance, we next tested if altered *P. mendocina* preference upon GRb0427 training is inherited for multiple generations. Indeed, F2 progeny of GRb0427-trained animals continue to exhibit reduced attraction to *P. mendocina* (**Fig 10F**).

Taken together, our results suggest Pv1 small RNA-mediated learned avoidance induces both avoidance of pathogenic *P. vranovensis* but also the avoidance of beneficial *P. mendocina* bacteria that may be nutritious food source for *C. elegans*. Therefore, it may be beneficial to reverse this memory after a few generations, when the worms may no longer encounter the original pathogen.

## Discussion

*C. elegans'* wild environments are rife with diverse bacterial species, and recent studies have provided comprehensive descriptions of *C. elegans'* natural microbial environments [43,44,47]. The effects of these bacterial species on *C. elegans'* nervous system and behavior have been largely unexplored, and whether these bacteria can exert transgenerational effects on worms was previously unknown. Here we show that a small RNA from *Pseudomonas vranovensis* GRb0427, a pathogenic bacterial species found in *C. elegans'* natural environment, induces learned avoidance in worms. This learned avoidance persists for four generations, similar to the learned avoidance induced by the P11 small RNA expressed in the clinical isolate of *P. aeruginosa*, PA14. This wild bacterial species, *P. vranovensis*, however, does not express the P11 small RNA; instead, it uses a distinct small RNA, which we identified and named Pv1, to induce avoidance and its transgenerational inheritance. The Pv1 small RNA, like P11, regulates avoidance by targeting the neuronal gene *maco-1* –but in a different exon—and subsequently upregulating the expression of the TGF-beta ligand *daf-7*, leading to the switch from attraction to avoidance (**Fig 9E**). Our results also indicate that the RNAi pathway is required for both the PA14/P11 and *P. vranovensis*/Pv1 mechanisms of bacterial sRNA-induced avoidance, as SID-2, DCR-1, and SID-1, as well as the 22G Argonaute, HRDE-1, are all required for Pv1-induced avoidance.

Our study provides an example of small RNA-mediated cross-kingdom signaling between bacteria and their animal hosts, adding to the diverse instances of RNA-based trans-kingdom signaling [23,69,70]. The identification of Pv1 expands the repertoire of bacterial small RNAs that can affect host physiology, and begins to identify common principles of bacterial small RNA-mediated regulation of *C. elegans* behavior. Both Pv1 and P11 target *maco-1*, a conserved ER-localized calcium regulator [62], indicating that the sRNA/*maco-1*/*daf-7* axis may be important for regulation by multiple bacterial small RNAs. Pv1, like P11, has a short perfect match to its putative target, *maco-1*, indicating that bacterial small RNAs require perfect sequence complementarity for sRNA-mediated effects; moreover, these matches are present in

stem loops of predicted secondary structures of these sRNAs (**Fig 8B**). However, Pv1 and P11 contain sequence matches to different exons of the *maco-1* gene sequence, suggesting that the P11-matched region of *maco-1* is not the key element. Although P11 and Pv1 are similar in secondary structure and length, they have very low sequence homology, suggesting that these two small RNAs likely evolved independently. The odds of a *Pseudomonas* small RNA having a perfect 16 nt match to an exon of *maco-1* by chance are low (less than 0.03, which perhaps is an overestimate; see Methods). Therefore, although the 16 nucleotide length is short, the perfect match requirement between a bacterial small RNA and a potential *C. elegans* transcript may sharply limit the possibilities. The fact that the *maco-1* sequence matches are present in predicted stem loops may also suggest that there may be convergent evolution towards a particular length or secondary structure. This supports the idea that several other bacterial sRNA-based regulation systems exist that may target the same or distinct *C. elegans* genes and may be yet to be discovered. Characterizing sRNAs from additional bacterial species that induce heritable behaviors in worms may help define molecular and structural features of sRNAs that can induce learning and inheritance, and may even allow us to predict which naturally existing exogenous sRNAs, including sRNAs from animal gut microbiomes, have the potential to induce heritable effects.

These findings establish bacterial small RNA-mediated regulation of host physiology as a phenomenon that may occur in the wild and in *C. elegans'* interaction with multiple different bacterial species. In fact, our data indicate that small RNAs may play a significant role in the neuronal response of *C. elegans* to wild bacteria: the behavioral response to GRb0427 is almost entirely mediated by small RNAs. In fact, the secondary metabolite-induced neuronal response observed upon *Pseudomonas aeruginosa* PA14 exposure (*daf-7* induction in ASJ neurons) [27,59] is not observed upon GRb0427 exposure, and the innate immunity gene *irg-1* [71] is not greatly induced, either, suggesting that the sRNA-mediated response, rather than classical innate immunity pathways, may be the major mechanism of avoidance of this pathogen. The sRNA-mediated learned response is also mechanistically distinct from a previously-described intergenerational effect on progeny (embryo) survival upon *P. vranovensis* exposure, which does not appear to use bacterial small RNAs or continue beyond the F1 generation [8], or a P0 avoidance effect induced by *O. vermis* [58]. Nor is there any overlap with the general response to pathogens [72], which is mediated by translational inhibition; for example, *vhp-1*, which plays a key role in the general response, has the opposite effect on pathogen avoidance [23]. The transgenerational effect we observe is species-specific; in fact, of the pathogenic species we tested here (Fig 1), and in previous work (*S. marcescens* [27]), only PA14 and GRb0427, two *Pseudomonas* pathogens, induce transgenerational inheritance of pathogen avoidance. Although not every *Pseudomonas* species induces avoidance (the beneficial *P. mendocina* does not, for example), of the pathogens we have tested, only *Pseudomonas* pathogens seem to induce a small RNA-mediated transgenerational avoidance response. Further work will determine how broadly *C. elegans* employs this mechanism to avoid *Pseudomonas* pathogens.

Transgenerational inheritance of learned avoidance of both PA14 and GRb0427 lasts for four generations, suggesting that there may be some benefit to "forgetting" learned avoidance of a pathogen. Here we ruled out temperature-dependent changes in pathogenicity and resulting fluctuations in Pv1 expression as factors. Instead, our results suggest that avoidance of pathogenic *P. vranovensis* may also cause (mistaken) avoidance of beneficial *Pseudomonas* species, such as the non-pathogenic food source, *P. mendocina*. Taken together, our results suggest that in the wild, Pv1 small RNA-mediated learned avoidance may protect worms and their progeny from the pathogenic *Pseudomonas*, but since this might also result in avoidance of beneficial bacteria, it may be advantageous to reverse this memory after a few generations, when the worms may no longer encounter the original pathogen. Reversal of this memory

may have evolved to protect worms from maladaptive avoidance of other beneficial bacterial species once the pathogen threat has passed. Since most bacterial families in *C. elegans'* microbiome include beneficial as well as detrimental members that may present similar odors to the worms, reversible transgenerational responses to multiple other bacterial species may have evolved as an adaptive strategy.

Together, our results suggest that the "interpretation" of bacterial small RNA signals may in fact be one reason that *C. elegans* developed such robust RNA interference mechanisms: in addition to endogenous RNAi systems used to silence errant germline transcripts [73–75], mechanisms to regulate and process exogenous sRNAs from the worm's environment and bacterial food sources may be critical for avoidance of pathogens in *C. elegans'* environment. The wild *C. elegans* strain JU1580 has a mutation in the viral RNAi Dicer-like homolog, *drh-1*, disabling the viral RNA interference pathway; these results suggest that *C. elegans* detects wild bacterial sRNA through the canonical RNAi pathway, not through the viral pathway. Together, our results suggest that environmental sRNAs–that is, sRNAs produced in the bacteria that live in *C. elegans'* habitat–can be taken up and processed by *C. elegans'* RNA interference pathway. Considering the worms' requirement for bacteria as its food and the broad spectrum of microbes in their environment, the ability to interpret bacterial sRNA signals may be a powerful mechanism of adaptive immunity. If *C. elegans* constantly surveys its bacterial environment, but also needs to cease avoidance after a few generations to prevent accidentally avoiding beneficial food sources, the bacterial small RNA/RNA interference pathway may provide an ideal adaptable, resettable response mechanism. More examples will be necessary to determine whether the system is restricted to *Pseudomonad* species and small RNAs that target *maco-1*, as well as to better define the characteristics of key bacterial small RNAs.

Our work provides proof of principle that transgenerational inheritance of learned avoidance is likely to benefit *C. elegans* in the wild. Although bacterial small molecules have been widely implicated in bacteria-host interactions [76], the potential roles of bacterial small RNAs in regulating host physiology are largely understudied. Identifying different bacterial small RNAs that can interact with host organ systems will lead to a better understanding of this new dimension of bacteria-host interactions.

## Methods

### Resource availability

Further information and requests for resources and reagents should be directed to and will be fulfilled by Coleen T. Murphy (ctmurphy@princeton.edu).

### Materials availability

Bacterial and *C. elegans* strains generated in this study are available on request.

### Experimental model and subject details

**Bacterial strains.**   The GRb0427 and Jub38 strains were a generous gift from Dr. Buck Samuel's lab. OP50 was obtained from the C.G.C. The PA14 strain was a gift from Prof. Zemer Gitai's lab. Control (L4440) and *maco-1* RNAi were obtained from the Ahringer RNAi library, and the respective sequences were verified. MSPm1, Jub66, Myb10, Myb71, CEN2ent1, Jub19, and Bigb0170 are part of the CeMbio [50] collection and was obtained from CGC.

### Engineered bacterial strains

**IntReg.**   *E. coli* expressing the GRb0427 347-nt intergenic region containing the 16-nucleotide match to *maco-1* was constructed by Gibson Assembly. The entire intergenic region was cloned out of GRb0427 and ligated into pZE27GFP using a double restriction digestion to open the plasmid and a single fragment Gibson assembly to ligate. This plasmid was then transformed into MG1655 *E. coli* using a standard transformation protocol.

**GRb0427Δ16.**   The deletion of the 16-nt region of Pv1 (that matches to *C. elegans maco-1*) from the GRb0427 genome was constructed by two-step allelic exchange using plasmid pEXG2. Briefly, ~600 bp fragments directly upstream and downstream of Pv1 sequence were amplified from GRb0427 gDNA using primer pairs (Pv1-KO1, Pv1-KO2) and (Pv1-KO3, Pv1-KO4), respectively. Overlap-extension PCR, with primer pair (Pv1-KO1, Pv1-KO4), was used to fuse together the upstream and downstream fragments. The final fragment was cloned into pEXG2, which was PCR linearized with primer pair (pEGX2-Lin1, pEGX2-seq2). The pEXG2 plasmid was integrated into GRb0427 by conjugation from donor strain *E. coli* s17 and exconjugants were selected on gentamycin 30 μg/mL and irgasan 100 μg/mL. Mutants of interest were counterselected on 15% sucrose and the proper deletion was confirmed via PCR, using primers (Pv1-seq1, Pv1-seq2), and sequencing of PCR products.

Primer details:

Pv1-KO1—CGCACCCGTGGAAATTAATTGCTTCAGTGAAGGGCGG
Pv1-KO2—TTCAGCATGCTTGCGGCTCGAGCACAGAAAGCAGATTAAATATGCGC
Pv1-KO3—CTCGAGCCGCAAGCATGCTGAATTTTAGCCGTACCGAACAAGC
Pv1-KO4—CCGGAAGCATAAATGTAAGCGTCCTTGTCGGGGC
pEGX2-Lin1—CTTTACATTTATGCTTCCGGCTCGTA
pEGX2-Lin2—AATTAATTTCCACGGGTGCGCATG

**Pv1.**   *E. coli* expressing the Pv1 small RNA was constructed in a similar method, using primers specific to the Pv1 region:

Pv1-seq1—CGCACCCGTGGAAATTAATTGCTTCAGTGAAGGGCGG
Pv1-seq2 –CCGGAAGCATAAATGTAAGCGTCCTTGTCGGGGC

The region encoding Pv1 was cloned out of GRb0427 and ligated into pZE27GFP using a double restriction digestion to open the plasmid and a single fragment Gibson assembly to ligate. This plasmid was then transformed into MG1655 *E. coli* using a standard transformation protocol.

**Pv1 mismatch.**   *E. coli* expressing the Pv1 small RNA with disrupted *maco-1* sequence homology, but with intact secondary structure, was constructed using the Pv1-expressing plasmid. Four mismatches were introduced into the *maco-1* match sequence such that the sequence homology is disrupted but the secondary structure is maintained. The mismatches were introduced in the PCR primers, Gibson assembly was used for ligation, and the plasmid was transformed into MG1655 as previously described. The primers used for cloning the plasmid containing the Pv1 sequence with the mismatches are:

PV1 (sequence disrupted/secondary structure maintained) 5'
AGCTTACTGTGACGTTTGCTAATAAACTTTTAGCCGTACCGAACAAGCT
PV1 (sequence disrupted/secondary structure maintained) 3'
ATTAGCAAACGTCACAGTAAGCTGATAAAATATGCGCCCGTAGCTCAGCT

### *C. elegans* strains

The following strains were used in this paper: N2 (wild type), AU133: *agIs17[Pmyo-2::mCherry + Pirg-1::gfp]*, FK181: *ksIs2 [Pdaf-7p::gfp + rol-6(su1006)]*, CQ759: *maco-1(ok3165)*, CQ667: *Cer1(gk870313)*, HC196: *sid-1(qt9)*, YY11: *dcr-1(mg375) III*, CQ738: *hrde-1(tm1200)*, HC271:

*ccIs4251 [(pSAK2) Pmyo-3::GFP::LacZ::NLS + (pSAK4) Pmyo-3::mitochondrial GFP + dpy-20 (+)] I; qtIs3 [Pmyo-2::GFP dsRNA hairpin] sid-2(qt42) III; mIs11 [Pmyo-2::GFP + Ppes-10::GFP + gut-promoter::GFP] IV*, and JU1580 and ED3040 (wild type, natural *C. elegans* isolates).

## Cultivation of bacterial strains

*E. coli* OP50, *P. vranovensis* GRb0427, *P. vranovensis* GRb0427 Δ16 nt, *Raoultella sp*. Jub38, the CeMbio strains, and *P. aeruginosa* PA14 were grown in LB (10 g/l tryptone + 5 g/l yeast extract + 10 g/l NaCl in distilled water) in a shaker at 250 rpm. *E. coli* expressing the Pv1 sRNA were grown in LB supplemented with 50 μg/mL Kanamycin. RNAi bacteria were grown in LB supplemented with 100 μg/ml carbenicillin and 12.5 μg/ml tetracycline.

## General maintenance of C. elegans strains

Worm strains were maintained at 20˚C on high-growth medium (HGM) plates (3 g/l NaCl, 20 g/l bacto-peptone, 30 g/l bacto-agar in distilled water, with 4 mL/L cholesterol (5 mg/mL in ethanol), 1 mL/L 1 M CaCl$_2$, 1 mL/L 1 M MgSO$_4$ and 25 mL/L 1 M KPO$_4$ buffer (pH 6.0) added to molten agar after autoclaving) on *E. coli* OP50.

## Method details

### Training plate preparation

Training plates were prepared by pipetting 800 uL of bacteria onto NGM (3 g/L NaCl, 2.5 g/L Bacto-peptone, 17 g/L Bacto-agar in distilled water, with 1 mL/L cholesterol (5 mg/mL in ethanol), 1 mL/L 1M CaCl2, 1 mL/L 1M MgSO4, and 25 mL/L 1M potassium phosphate buffer (pH 6.0) added to molten agar after autoclaving) or HG plates. Pathogenic bacteria such as PA14 or GRb0427 were prepared on NGM plates to avoid overgrowth, while sRNA producing MG1655 were prepared on HG plates. For sRNA training, 200 μl of OP50 was spotted in the center of a 6-cm NGM plate. Plates were stored at 25˚C for 48hrs. RNAi bacteria were prepared on HG plates (supplemented with 1 mL/L 1M IPTG, and 1 mL/L 100 mg/mL carbenicillin) and kept at room temperature for 48 hrs. Plates were then taken out of the incubator and allowed to cool to room temperature before moving animals onto them.

### Bacterial choice assay plate preparation

Overnight bacterial cultures were diluted in LB to an OD$_{600}$ = 0.5, and 25 μl of each bacterial suspension was spotted onto one side of a 60-mm NGM plate to make bacterial choice assay plates. These plates were incubated for 2 days at 25˚C.

### Preparation of bacteria for small RNA isolation

GRb0427 and *E. coli* OP50 bacteria were cultured for 16 hours overnight. 1 ml of either bacterial culture was diluted to an OD = 0.5, plated on 100mm NGM plates and grown at 25˚C for 48 hours. For 15˚C GRb0427 small RNA isolation, overnight (37˚C) cultures of GRb0427 bacteria were centrifuged for 10 min at 5,000*g*. The supernatant was removed, and the remaining pellet was resuspended in 5 ml of fresh LB. Washed bacteria were used to inoculate (1:500) fresh LB to grow at 15˚C for 36 hrs. 1 ml of this bacterial culture was then diluted to an OD = 0.5, plated on 100mm NGM plates and grown at 15˚C for 48 hours.

Bacterial lawns were collected from the surface of the plates using a cell scraper. 1 ml of M9 buffer was applied to the surface of the bacterial lawn, and the bacterial suspension obtained by scraping was transferred to a 15-ml conical tube. *E. coli* OP50 from 15 plates, or GRb0427 from 10 plates were pooled in each tube and pelleted at 4500*g* for 8 min. The supernatant was

discarded, and the bacterial pellet was resuspended in 900 uL of Trizol LS for every 100 µl of bacterial pellet. The pellet was resuspended by vortexing and the tubes containing the bacterial pellet were frozen at −80˚C.

## Bacterial small RNA isolation

To isolate small RNA, bacterial pellets-Trizol suspensions were first incubated at 65˚C for 10 min with occasional vortexing. Debris were pelleted at $4500g$ for 5 min. The supernatant was transferred to 1.5 mL tubes (1 mL in each tube) and 200uL chloroform was added. Samples were mixed by inverting and centrifuged at $12,000g$ at 4˚C for 10 min. The aqueous phase obtained was used as input for small RNA extraction using the mirVana miRNA isolation kit. Extraction was done as per the manufacturer's instructions for small RNA (<200 nt) isolation. Purified small RNA was used immediately in aversive learning assays or for sequencing, or frozen at −80˚C until further use.

## Training *C. elegans* on bacterial lawns and small RNAs

Wild-type N2 animals were synchronized by bleaching and grown until larval stage 4 (L4) on standard HG plates seeded with OP50. At L4 stage, they were transferred to training plates.

After 48 hrs at 25˚C the training plates were left on the bench top for 30 mins to allow them to reach room temperature. For sRNA training, 100 ug of sRNA was added to the OP50 spot on the sRNA training plates. For RNAi, 200 µL of 0.1M IPTG was spotted onto seeded RNAi plates and left to dry at room temperature before adding worms. Larval stage 4 (L4) worms were washed off plates using M9 and left to pellet on the bench top for 2–3 min. Then, 5 µl of worms were placed onto sRNA-spotted training plates, 10 µl onto OP50 plates, or 20 µl onto RNAi plates, *E. coli* expressing Pv1, GRb0427, or GRb0427Δ16 training plates. Worms were incubated on training plates at 20˚C in separate containers for 24 hours. After 24 hours, worms were washed off plates using M9 and washed an additional 2–3 times to remove excess bacteria. Trained worms were tested in the aversive learning assay.

## Aversive learning assay

On the day of the assay, bacterial choice assay plates were left at room temperature for 1 h before use. To start the assay, 1 µl of 1 M sodium azide was spotted onto each respective bacteria spot to be used as a paralyzing agent during choice assay. Worms were then washed off training plates in M9, allowed to pellet by gravity, and washed 2–3 additional times in M9. Using a wide orifice pipet tip, 5 µl of worms were spotted at the bottom of the assay plate, midway between the bacterial lawns. The assay plates were incubated at room temperature for 1 h. After that, the number of worms on each bacterial spot were counted. Plating a large number of worms (>200) on choice assay plates was avoided, because the worms clump at bacterial spots, making it difficult to distinguish individual worms during counting, and also because high densities of worms can alter behavior.

For experiments testing behavior of the F1 generation, day 1 worms from parental ($P_0$) training were bleached and eggs were placed onto HG plates and left for 3 days at 20˚C. After 3 days, the F1 worms (Day 1–72 hours) were washed off HG plates with M9. Some of the pooled worms were subjected to the aversive learning assay, and the remaining worms were bleached to obtain eggs. The eggs were then placed onto HG plates, which were left at 20˚C. After 3 days the F2 progeny were tested, and the same steps were followed for subsequent progeny generations.

### Annotation of the GRb0427 genome

The GRb0427 genome was obtained from [8] and run through an annotation pipeline in python. This pipeline searches for all open reading frames across the genome on both strands and makes a temporary gene list. It then filters genes by both size and overlap with other genes to give a final predicted gene list. Our pipeline predicted 4952 genes in the genome. Multiple genes were then selected and run through NCBI BLAST to confirm their identities.

After identifying the genes in the genome, we used Standalone BLAST, specifically, BLAST-nShort, to identify regions of homology between *maco-1* and the GRb0427 genome. This yielded 5 hits: 4 hits of 16 nucleotides and 1 of 20 nucleotides. Next, these hits were run through a python function to determine if they lie in an intergenic region, and this filtered our list down to a singular 16 nucleotide hit. This hit lies within a predicted 347nt intergenic region.

### Bacterial small RNA sequencing

Prior to sRNA sequencing, each sample of GRb0427 sRNA was tested for *C. elegans* behavior. The size distribution of sRNA samples was examined on a Bioanalyzer 2100 using RNA 6000 Pico chip (Agilent Technologies). The sRNA sequencing protocol was similar to the protocol used in [23]. Briefly, around 300 ng of sRNA from each sample was first treated with RNA 5′ pyrophosphohydrolase (New England Biolabs) at 37°C for 30 min, then converted to Illumina sequencing libraries using the PrepX RNA-seq library preparation protocol on the automated Apollo 324 NGS Library Prep System (Takara Bio). The treated RNA samples were ligated to two different adapters at each end, then reverse-transcribed to cDNA and amplified by PCR using different barcoded primers. The libraries were examined on Bioanalyzer DNA High Sensitivity chips (Agilent) for size distribution, quantified by Qubit fluorometer (Invitrogen), and then pooled at equal molar amount and sequenced on Illumina NovaSeq 6000 S Prime flowcell as single-end 122-nt reads. The pass-filter reads were used for further analysis.

### Bacterial small RNA sequencing data analysis

Four replicates of GRb0427 small RNA and three replicates of *E. coli* OP50 small RNA were sequenced. Reads were mapped to the sequenced GRb0427 genome [8] using RNA STAR [77]. Default settings were used for the RNA STAR mapping. The resulting BAM files were then loaded into IGV genome browser [78] for analysis of the intergenic region containing the 16-nucleotide sequence match to *maco-1*. The peaks and read strands indicated that 3 small RNAs lie in the intergenic region of interest, and one spans the 16nt region of match to *maco-1*. Using the Sapphire promoter analysis software for *Pseudomonas* species [79], we found highly confident predicted promoters that were consistent with the sequencing data. Of the three small RNAs, one contains the 16 nt homology to *maco-1*, and this small RNA was named Pv1. The boundaries of Pv1 were determined based on the depletion of mapped reads at the same genomic positions across all four GRb0427 small RNA sequence datasets. sRNA data available at NCBI BioProject PRJNA1062118.

### Determination of the operon context of Pv1

We examined if Pv1 lies in a GRb0427 operon. Using the Operon Mapper operon detection software [80], we found that while Pv1 is flanked by operons for iron metabolism and sugar transport, Pv1 itself is not part of an operon.

## Pv1 structure prediction

The small RNA that has homology to *maco-1* was identified by our sequencing data to be 124 nucleotides in length. Using the mFold sRNA secondary structure prediction tool on the UNAFold webserver [81], we found that this RNA contains a long stem loop structure with the sequence match to *maco-1* beginning in a stem and ending after the turn of a loop. Interestingly P11's sequence match to *maco-1* has a similar secondary structure context.

## Imaging and image analysis

*daf-7p*::*gfp* images of OP50, PA14, and GRb0427 worms were taken on a Nikon Eclipse Ti microscope. Worms were prepared and treated as described in 'Worm preparation for training'. Worms were mounted on 2% agar pads on glass slides and immobilized using 1 mM levamisole. Z-stack multi-channel (DIC and GFP) images of day-1 adult GFP-transgenic worms were acquired at 60X magnification. Maximum intensity projections of head neurons were built using Fiji. Quantification of mean fluorescent intensity was done using NIS Elements software. Average pixel intensity was measured in each worm by drawing a Bezier outline of the neuron cell body for 2 ASI head neurons. For imaging of *daf-7p*::*gfp* in each generation, exposure times were adjusted to prevent oversaturation., For control and treatment groups imaged on the same day, the same exposure time and camera settings were used.

For *irg-1p*::*gfp* quantification, worms were prepared as described in 'Worm preparation for training' and imaged at 20× magnification on a Nikon A1 R confocal microscope. Image analysis was done with Fiji, where ROIs were drawn around each animal in the field of vision and mean intensity values for all regions of interest were recorded and plotted.

## *C. elegans* Survival assay

Survival assay on GRb0427 and GRb0427Δ16 lawns:

GRb0427 and GRb0427Δ16 bacteria were grown in liquid culture overnight (37˚C) and diluted 1:4 to an OD = 0.5. 750 µL of diluted GRb0427 or GRb0427Δ16 was spread to completely cover six 6-cm NGM plates for each bacterial genotype. Plates were incubated for 2 days at 25˚C to allow bacterial growth. Plates were equilibrated to 20˚C before adding worms (84 hours post-bleach) to plates. Survival assays were performed at 20˚C. The assay plates were counted every 6–9 h. Every 48h, worms were moved onto new plates.

Survival assay on 25˚C-grown and 15˚C-grown GRb0427 lawns:

Preparation of 25˚C-grown GRb0427 survival assay plates—Overnight (37˚C) cultures of GRb0427 bacteria were diluted in LB to ($OD_{600}$) = 0.5 and used to fully cover six 6-cm nematode growth medium (NGM) plates. The plates were incubated for 2 days at 25˚C.

Preparation of 15˚C-grown GRb0427 survival assay plates—Overnight (37˚C) cultures of GRb0427 bacteria were centrifuged for 10 min at 5,000g. The supernatant was removed, and the remaining pellet was resuspended in 5 ml of fresh LB. Washed bacteria were used to inoculate (1:500) fresh LB to grow at 15˚C for 36 hrs. Cultures were diluted in LB to an $OD_{600}$ = 0.5 and used to seed NGM plates. The plates were incubated at 15˚C for 2 days.

Assay procedure—Plates were equilibrated to 20˚C before adding Day 1 (72 hours post-bleach) worms to plates. Survival assays were performed at 20˚C. The assay plates were counted every 6–9 h. Every 24h, worms were moved onto new plates of 25˚C-grown and 15˚C-grown GRb0427 (prepared as described above).

## PCR detection of the Pv1 small RNA

Total RNA and small RNA were extracted from 25°C-grown and 15°C-grown GRb0427 using the mirVana miRNA isolation kit, and reverse-transcribed to DNA (using SuperScript III First Strand Synthesis System). A 62 bp region and an 88 bp region of Pv1 were identified using the following PCR primers:

Pv1(62bp)Fwd: CTGTGACGATTACAAATTAAC
Pv1(62 bp)Rev: GCCGTACCGAACAAG
Pv1(88 bp)Fwd: GCCTAGCACTGGTTAG
Pv1(88bp)Rev: CTGTGACGATTACAAATTAAC

## Quantification of *maco-1* gene expression by qPCR

Worms were trained for 24 hours on *E. coli* OP50 and GRb0427 (as described in the 'Training *C. elegans* on bacterial lawns and small RNAs' section). After training, the worms were collected in M9 and washed several times to remove excess bacteria. Worm pellets were crushed in liquid nitrogen and transferred to an appropriate volume of Trizol LS (100 μL of worm pellet in 900 μL of Trizol). Total RNA was extracted from sample-Trizol suspensions using chloroform extraction, isopropanol and ethanol precipitation, and cleanup using the RNeasy mini kit. cDNA was made from 1 ug of RNA using the Superscript III First-strand system for RT-PCR. The extracted cDNA was used as input for qPCR reactions using the Power SYBR green qPCR master mix and protocol and run on a Viia7 Real-time PCR system. The qPCR primers used are listed below:

*maco-1* Forward: GTGTCACGACAATTGCC
*maco-1* Reverse: CACATAGGTAGTGGCGAG
*act-1* Forward: GGCCCAATCCAAGAGAGGTATC [82]
*act-1* Reverse: CAACACGAAGCTCATTGTAGAAGG [82]

## Statistical analysis

Survival assays were assessed using Log-rank (Mantel-Cox) tests. For the comparison of choice indices between more than two genotypes, one-way ANOVA with Tukey's multiple comparisons test was used. For comparisons of choice indices between genotypes and between conditions (naïve vs learned), two-way ANOVA with Tukey's multiple comparisons test was used. Unpaired t tests were performed for comparisons between two groups. Experiments were repeated on separate days with separate populations, to confirm that results were reproducible. Prism 9 software was used for all statistical analyses.

## Perfect match calculation

There are approximately 500 small RNAs in *P. aeruginosa* [83] with an average length of 188 nt, constituting a total of ~86,000 16-nt windows within these small RNAs. The length of the *maco-1* coding sequence is approximately 2700 nucleotides, and so contains ~1,350 semi-independent 16-nt windows (allowing up to 90% overlap between neighboring 16-nt windows). The product of these two numbers is ~116,100,000 pairs of potentially matching windows. Dividing by $4^{16}$ possible 16-nt sequences yields an estimated probability of ~0.027. It should be noted that this number is likely an overestimate, as it assumes that the 86,000 windows in the small RNAs are independent of one another, which is not the case.

## Supporting information

**S1 Fig. 36-hour training of worms on Bigb0170 do not induce learned avoidance of Bigb0170.** Each dot represents an individual choice assay plate. Boxplots: center line, median;

box range, 25th–75th percentiles; whiskers denote minimum-maximum values; ns, not significant; Unpaired, two-tailed Student's t test.
(PDF)

**S2 Fig. F2 progeny of GRb0427-trained animals have higher *daf-7p::gfp* expression in the ASI sensory neurons (blue arrowheads), compared to F2 progeny of OP50-trained animals.** Scale bar = 10 μm.
(PDF)

**S3 Fig. (A)** 62 bp and 88 bp of Pv1 amplified by two different primer sets (set 1 and set 2) from total RNA pool extracted from 25˚C and 15˚C-grown GRb0427 (indicating expression of the Pv1 small RNA under both temperature conditions). **(B)** 62 bp and 88 bp of Pv1 amplified by the same primer sets as in (A) (also see Methods) from small RNA pool extracted from 25˚C and 15˚C-grown GRb0427. +RT and -RT indicate presence and absence respectively of the reverse transcriptase enzyme.
(PDF)

**S4 Fig. (A-C)** Naïve worms do not exhibit a preference towards either *P. vranovensis* GRb0427 or *P. mendocina* MSPm1 in a GRb0427-MSPm1 choice assay (A) while showing preference for GRb0427 and MSPm1 with respect to OP50 (B,C).
(PDF)

**S1 Data. Source data files for all figures.**
(XLSX)

## Acknowledgments

We thank Buck Samuel for sharing the GRb0427 and Jub38 bacterial strains, the *C. elegans* microbiome resource CeMbio for the remaining bacterial strains, and the *C. elegans* Genetics Center for worm strains; W. Wang, J. Arly Volmar, and J. Miller (Genomics Core Facility, Princeton University); the Murphy lab for discussion and feedback; and J. Ashraf and W. Keyes for assistance.

## Author Contributions

**Conceptualization:** Titas Sengupta, Rachel Kaletsky, Rebecca S. Moore, Zemer Gitai, Coleen T. Murphy.

**Formal analysis:** Titas Sengupta.

**Funding acquisition:** Cameron Myhrvold, Coleen T. Murphy.

**Investigation:** Titas Sengupta, Jonathan St. Ange, Rachel Kaletsky, Rebecca S. Moore, Renee J. Seto, Jacob Marogi, Cameron Myhrvold.

**Methodology:** Titas Sengupta, Rachel Kaletsky, Rebecca S. Moore, Jacob Marogi.

**Project administration:** Zemer Gitai, Coleen T. Murphy.

**Software:** Jonathan St. Ange.

**Supervision:** Cameron Myhrvold, Zemer Gitai, Coleen T. Murphy.

**Validation:** Titas Sengupta, Renee J. Seto.

**Writing – original draft:** Titas Sengupta, Jonathan St. Ange, Rachel Kaletsky.

**Writing – review & editing:** Cameron Myhrvold, Zemer Gitai, Coleen T. Murphy.

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
