## [Decision Letter · Decision Letter 0]

30 Nov 2023

Dear Coleen,

Thank you very much for submitting your Research Article entitled 'A natural bacterial pathogen of C. elegans uses a small RNA to induce transgenerational inheritance of learned avoidance' to PLOS Genetics.

The manuscript was fully evaluated at the editorial level and by three independent peer reviewers. The reviewers appreciated the attention to an important problem, but raised some concerns about the current manuscript. Based on the reviews, we will not be able to accept this version of the manuscript, but we would be willing to review a revised version. We cannot, of course, promise publication at that time.

The external peer reviews were split.  Reviewers #2 and #3 were very enthusiastic, commenting that the central conclusions are exciting and that the manuscript was well written.  Reviewer #2 had only a minor formatting suggestion for Figure 1.  Reviewer #3 had more extensive comments, but they should all be relatively straight forward to address.  Reviewer #1 was the most critical, wavering between recommending Major Revision or Rejection (though they acknowledge that the central claims of the manuscript are important advances). I think it will be important to address the concerns in their “mechanism” critiques. Addressing the concerns described in the “Ecological Relevance” critiques would (as far as I understand) require field pathology work that is in my opinion probably beyond the scope of this study.  Instead, I think this could be addressed by, as the reviewer suggests, toning the language down and presenting the ecological relevance as a conjectural hypothesis that will need to be confirmed by follow-on natural history studies.  Some of the concerns in the “Rationale for limited transgenerational inheritance” critiques are also echoed by Reviewer #3.  If these could be at least partially addressed by additional experimentation (both Reviewer #1 and #3 offer some potential directions), I think it would significantly strengthen the paper.  At the very least, toning the language down here would also help (although I disagree, with all due respect, with the reviewer that demonstrating this in the lab, as opposed to the wild, is insufficient).

If you decide to revise the manuscript for further consideration at PLOS Genetics, please aim to resubmit within the next 60 days, unless it will take extra time to address the concerns of the reviewers, in which case we would appreciate an expected resubmission date by email to plosgenetics@plos.org.

We are sorry that we cannot be more positive about your manuscript at this stage. Please do not hesitate to contact us if you have any concerns or questions.

Yours sincerely,

Gregory P. Copenhaver

Editor-in-Chief

PLOS Genetics

Gregory Barsh

Editor-in-Chief

PLOS Genetics

Reviewer's Responses to Questions

**Comments to the Authors:**

Reviewer #1: C. elegans exhibits a remarkable tendency for transgenerational epigenetic inheritance of RNAi-mediated silencing. Whilst mechanistically this is fairly well characterised, the ecological context of this system, if indeed it exists, is largely unknown. In the current manuscript the authors build on several previous papers from the same team in which they purport to explain transgenerational RNAi may be an evolved response to pathogenic bacteria encountered by C. elegans. Previously their model was based on experiments with P. aeruginosa, which is not a natural pathogen of C. elegans. In the current manuscript they argue that a natural pathogen of C. elegans, P. vavroensis (previously described) has the same effect.

The manuscript presents two claims which, I believe, the authors argue are important conceptual advances. First, they argue that the fact that P. vavroensis is a “natural” pathogen of C. elegans supports the ecological role of the transgenerational epigenetic inheritance that they observe in the laboratory. Second, they argue that, because P. vavroensis exposure can lead to C. elegans avoiding non-pathogenic food sources, their experiments “explain” why the transgenerational epigenetic inheritance has evolved to only last a limited number of generations. They also present a mechanism whereby the bacteria can induce a behavioural response that is inherited transgenerationally. If supported by strong data these advances would indeed be important. However, I don’t think that the experimental results support the strong claims made in the manuscript. Whilst the gaps in the mechanistic understanding could be filled by experiments (and I suggest some of the ways in which this could be done), the way in which the manuscript attempts to link their laboratory observations to ecological scenarios must be toned down or removed as they are extremely speculative and could never be justified by the kind of laboratory experiments that the authors put forward.

Below, I outline these major issues in more detail.

Major issues

1. Mechanism

The authors suggest that a small non-coding RNA produced by the bacteria (i) is taken up by the worms (ii), processed by the RNAi pathway (iii) and triggers silencing of a specific genes maco-1 (iv) leading to the behavioural response that is inherited for 4 generations (v).

More evidence is required for each of these steps.

i) The authors present no solid evidence that a small non-coding RNA is actually produced. They claim to have done small RNA sequencing of the bacteria but then just present (in Fig 7A) a structural model of a small RNA that they claim to have discovered. What is the evidence for this? At a minimum there needs to be some presentation of the genome-wide small RNA sequencing data from the bacteria, showing that small RNA is recovered from this region far above other non-coding regions of the genome, and that the start and end points of this match their predicted structure. Moreover, it would be important to verify that this specific, highly structured, prediction actually corresponds to the RNA that is produced- some kind of RACE perhaps where they can map the 5’ and 3’ terminus of the RNA produced by the bacteria.

Ii/iii)

There is no small RNA sequencing data of worms that have eaten the bacteria presented, so there’s no evidence that the non-coding RNA produced by the bacteria is actually processed by the RNAi pathway (i.e. does it get chopped up by dicer). The authors should perform this experiment to show that short dsRNA really are produced from this sequence (see also point iii). Furthermore, the authors should perform the experiment in a sid-2 mutant as this would not be able to take up dsRNA from the bacteria and so, if the authors are correct, would not trigger the inheritance response.

iv) For silencing to occur 22G-RNAs need to be induced against the maco-1 gene as a result of the limited sequence identity (16bp) in the non-coding bacterial RNA. I find this very hard to believe – if 16bp were sufficient to trigger silencing then C. elegans RNAI by feeding would be riddled with off-target effects and would be impossible to interpret. In RNAi by feeding much longer regions of sequence similarity ~100-200bp as a minimum are required. Therefore, the authors need to verify that they really are seeing maco-1 22G-RNAs triggered by feeding with P. vavroensis and the 347bp region containing the supposed non-coding RNA. The sequencing described in ii) would be able to recover these if the authors use RppH to remove the 5’ triphosphate on 22G-RNAs.

v) P. vavroensis-induced gene expression changes should, if the authors’ model is correct, last for 4 generations. There is no data presented to support whether maco-1 is altered in expression for 4 generations, F1, F2, F3, f4- qPCR in Fig 5E is only in P0- and then goes back to normal. This data is important and if the result of this shows that maco-1 does not show sustained downregulation, the authors model to explain their findings would be seriously undermined. The authors also mention that an independent dataset (Burton et al., 2020) showed downregulation of maco-1 but they do not specify whether this is in P0, F1 or F2 animals- this data is all in the cited dataset so they should quote figures for all of these generational time points. Again, it would strongly undermine their claims if the effect is only in P0.

2. Ecological relevance

The main reason why this study would be a substantial advance over the authors’ previously published work on this topic would be that this is an ecologically relevant interaction. However, there is insufficient data to conclude this point. All we know is that P. vavroensis can be found with C. elegans in the wild and that, in the lab, it is pathogenic. We do not know that it is a pathogen in the wild and, to my knowledge, P. vavroensis has only been sampled once so we have no knowledge of how frequently C. elegans is likely to be exposed to it. Unless the authors can provide some evidence that this is a frequently observed interaction, their claim that this is somehow an evolved response is too strong and should either be toned down or removed. This then challenges the novelty of their observations since they have already published that Pseudomonas derived small RNAs from other species have an effect on behaviour, inherited transgenerationally, in the lab.

3. Rationale for limited transgenerational inheritance

The authors make the startling claim that the avoidance response may be limited because of the possibility that beneficial bacteria are avoided incorrectly. This goes way beyond what they actually show in the manuscript. Their experiment shows only that the exposure to pathogen triggers avoidance of a non-pathogenic bacteria as well but they do not show that there is a fitness cost associated with this even in the lab, let alone the wild. It would be possible to test this in the lab: a competition experiment between inheritance competent and inheritance incompetent strains (i.e. a WT vs a hrde-1 mutant) grown on a mixture of pathogenic and non-pathogenic bacteria or a fluctuating environment (pathogenic vs non-pathogenic in different generations). See which strain wins- if the inheritance incompetent strain wins then there may be some support for the authors’ assertation. Even this, though, would not really be sufficient to address whether this is actually relevant in the wild. Thus I think this entire section (Fig 8) should be toned down to report simply what the experiment shows, which does not involve anything about the importance of “forgetting” the avoidance response, only that the response itself has some cross-species effects that might not be directly useful.

Reviewer #2: The manuscript by Sengupta et al builds on their previous work showing that the PA14 small RNA of the human pathogen Pseudomonas aeruginosa is required for transgenerational inheritance of learned pathogenic avoidance in C. elegans. In this current study, authors seek to determine if a similar mechanism operates in the worm’s naturally occurring microbial environment. Authors survey nine bacterial species for the ability to induce learned avoidance and find one pathogenic species – Pseudomonas vranovenis/GRb0427 – that upregulates the daf-7p::gfp reporter in ASI neurons, similar to PA14. Unlike PA14, GRb0427 does not activate daf-7 expression in another set of neurons and does not robustly activate the innate immune response, suggesting a shared and distinct mechanism. The GRb0427 avoidance response is transmitted for four generations and requires a small RNA produced by GRb0427. Authors use a clever and rigorous strategy to identify the intergenic region Pv1 RNA, which has a 16-nt match to the maco-1 genes. (Whilst Pv1 and PA14 both have homology to maco-1, the matches are in different exons.) Pv1 and these 16nt are sufficient and necessary for transgenerational inheritance. Authors look at other Pseudomonas species and find that Pv1 induces avoidance of the beneficial bacterial MSPm1 and propose that “forgetting” by the 5th generation may protect C. elegans from maladaptive avoidance. These findings are exciting and important – this work and Kaletsky et al (2020) indicate that small RNA-mediated transgenerational inheritance may be a broad mechanism that C. elegans uses to avoid Pseudomonas pathogens. This manuscript is exceptionally well written, the figures are convincing and easy to follow, and the sophisticated genetic/genomic analysis combined are perfect for the readership of PLoS Genetics.

I really have no criticisms or concerns. My only suggestion is to rearrange Figure 1 so that panels E, G, and D are directly above panels L, M, and N – this will make it easier to compare P0 with the F1 generation.

Reviewer #3: In the manuscript “A natural bacterial pathogen of C. elegans uses small RNA to induce transgenerational inheritance of learned avoidance” Sengupta T et al perform a series of thorough experiments to investigate whether bacteria that C. elegans are naturally exposed to elicit the same transgenerational memory that they had previously demonstrated for an alternative bacteria that is pathogenic to humans (Pseudomonas aeruginosa). They showed that another Pseudomonas strain, Pseudomonas vranovensis, also elicits an epigenetic memory. They show this is dependent on the small RNAs isolated from the P. vranovensis. Furthermore they demonstrate that a specific small RNA which targets the same gene in C. elegans, maco-1, which they had demonstrated was targeted by P. aeruginosa small RNAs to elicit epigenetic memory, is also at play in the second Pseudomonas strain. They demonstrate that this small RNA is necessary and sufficient to induce the epigenetic memory by knocking it out in the Pseudomonas and expressing it in other types of bacteria. All in all I thought this is a very thorough and well done paper and mostly minor corrections are required for publication.

Major points

1) Are there beneficial memories with regards to the good bacteria? It seems like all the types of bacteria here are inducing an avoidance (even the beneficial one (Myb71)). Why are they learning to avoid the beneficial bacteria (Myb71) the same as they are the detrimental bacteria (GRb0427) in the P0 generation? This just seems like avoidance of foreign rather than avoidance of detrimental. Unless some beneficial memory can be demonstrated or if already published referenced then a restructuring of the framing of the manuscript is necessary. Can the small RNA (IntReg or Pv1) be added to a bacteria that produces beneficial memories (not just preferential bacteria) and change it to a negative multigenerational memory?

2) Need to explain what MACO-1 is early before introducing dependence on it in Fig. 5. What is the protein, what is its function, why is it important for memory etc… Why is a microtubule binding protein at all important for transmitting memory? Is this protein binding to anything differently when exposed to a memory inducing bacteria than regular bacteria? Is it differentially modified? Some additional experimental reason for the importance of this protein would greatly strengthen the paper. This didn't seem to be provided in last manuscript or this one and the readers are left wondering why this particular protein is important here.

Minor points

1) First sentence of summary ”its” is not appropriate here.

2) Page 5 duplicate citations of 47 and 50.

3) In Figure 1 it would be helpful to put an indicator on the actual figure of which strain is pathogenic, which are beneficial and which are positive/neutral depending on context. Or at least group them as you introduced them in the text. It’s also confusing to have multiple different nomenclature for the same thing (Jub10 and Stenotrophomonas maltophilia). Just pick one nomenclature and stick with that (I would suggest the scientific one rather than the jargony abbreviation which can be put in the methods).

4) Do the other strains exhibit learning after longer exposure to the bacteria or is it only those 3 at any length of training? And do those other strains not have any small RNAs at all homologous to maco-1?

5) Scale bars should be on every image not just on 1 of multiple images in a series (Fig 2A, C, E)

6) What is the magnitude in increase in daf-7 GFP relative to exposure to PA14? Is this same magnitude, dramatically less, dramatically more? Positive control for Fig 2D.

7) This sentence “This result is consistent with the daf-7p::gfp and irg-1p::gfp results (Figure 2E-F) suggesting that innate immunity pathways may not contribute significantly to C. elegans’ avoidance of P. vranovensis, but rather that the major pathway of avoidance in P0s is through the same pathway as in F1-F4.” Requires the reader to have extensive knowledge of your previous work. If you want to include this info need to summarize more briefly innate immune pathways etc from previous work and not assume reader has read your previous work.

8) I’m not sure what 3D provides that is different from 3C and same for 4E vs 4D. I’d say remove the second one as it is less informative. Or put to supplemental as they appear redundant.

9) I got a bit confused as to what the difference between the IntReg is and the Pv1? I thought the IntReg was the maco-1 homology region but then in the next paragraph and figure you say it is Pv1??

10) Figure 7D needs controls to show in that particular experiment choice is still working in your hands and that they weren’t flukes. (preference to OP50). Same for 8C.

11) The temperature experiments in Figure 8a+b seem like an offshoot and not germane to major point of manuscript. Move them to supplementary or better to just remove completely.

**Have all data underlying the figures and results presented in the manuscript been provided?**

Reviewer #1: **No: **I could not see a link to the repository (i.e. SRA or simiilar) containing their bacterial high throughput sequencing although the authors' statement implies that this is accessible.

Reviewer #2: Yes

Reviewer #3: Yes

PLOS authors have the option to publish the peer review history of their article (what does this mean?). If published, this will include your full peer review and any attached files.

Reviewer #1: No

Reviewer #2: No

Reviewer #3: No

---

## [Editor Report · Decision Letter 1]

9 Feb 2024

Dear Dr Murphy,

We are pleased to inform you that your manuscript entitled "A natural bacterial pathogen of C. elegans uses a small RNA to induce transgenerational inheritance of learned avoidance" has been editorially accepted for publication in PLOS Genetics. Congratulations!

Yours sincerely,

Gregory P. Copenhaver

Editor-in-Chief

PLOS Genetics

Gregory Barsh

Editor-in-Chief

PLOS Genetics

Comments from the reviewers (if applicable):

**Data Deposition**

http://datadryad.org/submit?journalID=pgenetics&manu=PGENETICS-D-23-01148R1

**Press Queries**

---

## [Editor Report · Acceptance letter]

4 Mar 2024

PGENETICS-D-23-01148R1 

A natural bacterial pathogen of <i>C. elegans<i> uses a small RNA to induce transgenerational inheritance of learned avoidance 

Dear Dr Murphy, 

We are pleased to inform you that your manuscript entitled "A natural bacterial pathogen of <i>C. elegans<i> uses a small RNA to induce transgenerational inheritance of learned avoidance" has been formally accepted for publication in PLOS Genetics! Your manuscript is now with our production department and you will be notified of the publication date in due course.

With kind regards,

Judit Kozma

PLOS Genetics

On behalf of:
